# ONLINE CONVEX OPTIMIZATION WITH PREDICTION THROUGH ACCELERATED GRADIENT DESCENT

## ABSTRACT

We study online convex optimization with predictions, where, at each time step $t$, predictions about the next $k$ steps are available, and with coupled costs over time steps, where the cost function at time step $t$ depends on the decisions made between time $t - a$ and time $t + b$ for some nonnegative integers $a, b$.

We provide a general recipe to run synchronous update in an asynchronous fashion that respects the sequential revelation of information. Combined with existing convergence results for convex optimization using inexact first-order oracle, we show that acceleration is possible in this framework, where the dynamic regret can be reduced by a factor of $(1 - O(\sqrt{\kappa}))^{\frac{k}{a+b}}$ through accelerated gradient descent, at a cost of an additive error term that depends on the prediction accuracy. This generalizes and improves the $(1 - \kappa/4)^k$ factor obtained by Li & Li (2020) for $a + b = 1$. Our algorithm also has smaller dependency on longer-term prediction error. Moreover, our algorithm is the first gradient based algorithm which, when the strong-convexity assumption is relaxed, constructs a solution whose regret decays at the rate of $O(1/k^2)$, at a cost of an additive error term that depends on the prediction accuracy.

## 1 INTRODUCTION

We study online convex optimization with coupled cost: at time step $t$, the cost function $f_t$ is a function of decisions $x_{(t-a):(t+b)}$, i.e., the decisions made in a window of length $a + b$ around $t$. This generalizes the well studied smoothed online convex optimization problem (Li & Li, 2020; Li et al., 2021; Goel & Wierman, 2019; Chen et al., 2018; Goel et al., 2019; Pan et al., 2022) where $f_t$ is the sum of a stage cost that depends only on current decision $x_t$, and a switching cost between $x_t$ and $x_{t-1}$. Following the setup of Li & Li (2020) and Li et al. (2021), we assume that the cost at time $t$ is parameterized by $\theta_t \in \Theta$, and the decision maker has potentially inexact predictions, $\widehat{\theta}_s^{(t)}$, about future $\theta_s$ ($s \geq t$). Online convex optimization with switching costs has been used in various settings such as online optimal control (Li et al., 2019), data center management (Lin et al., 2012), power systems (Kim & Giannakis, 2017), to name a few. Our history-dependent stage costs also echo a recent line of work on online convex optimization with memory (Anava et al., 2015; Kumar et al., 2023; Shi et al., 2020), which shows applications in online linear control (Agarwal et al., 2019), statistical arbitrage in finance, and time series prediction (Anava et al., 2015).

We focus on the following question proposed and studied in Li & Li (2020): *when making decisions online, how can one make the best use of predictions about future, while being robust to inaccuracy in the (long-term) predictions?*

Many methods have been proposed to incorporate predictions in online convex optimization: from optimization based methods such as RHC (Kwon & Pearson, 1977), AFHC (Lin et al., 2012) and CHC (Chen et al., 2016), which require solving optimization problems (exactly) at each iteration, to gradient based methods RHAG (Li et al., 2021) which converges at the optimal rate but requires exact prediction, and RHIG (Li & Li, 2020) which works with inexact prediction but suffers suboptimal convergence rate.

In this work, we propose the online Projected Gradient Method (online-PGM) and online Accelerated Gradient Method (online-AGM), which build upon variants of the well known (accelerated) gradient descent designed for inexact first order oracle (Bubeck, 2015; Devolder et al., 2013a;b). We show that

when $k$-step predictions of the objective functions and $k$-step look ahead initialization are available, online-AGM achieves the convergence rate $(1 - O(\sqrt{\kappa})^k)$ for $\kappa$-conditioned objective functions. In addition, the extra additive term in the dynamic regret due to prediction inaccuracy has smaller dependency on long-term prediction error, as compared to RHIG in Li & Li (2020). We also consider the case when the strong-convexity assumption is relaxed, and show that online-AGM constructs a solution whose regret decays at the rate of $O(1/k^2)$, plus additive terms due to prediction inaccuracy.

As a by-product, we formalize and generalize Li & Li (2020) and Li et al. (2021)'s "fill-the-table" approach to running (accelerated) gradient descent online, which might be of independent interest.

## 1.1 SETUP

We consider online convex optimization, where the loss at time $t$ depends on the decision $x_s$ in the window $s \in W_t = [\max(1, t - a), \min(t + b, T)]$ for some fixed $a, b \in \{0, 1, \ldots\}$, as well as a parameter $\theta_t \in \Theta$ in a parameter space. That is,

$$C(\mathbf{x}; \boldsymbol{\theta}^*) = \sum_{t=1}^{T} f_t(x_{W_t}; \theta_t^*). \tag{1}$$

At the beginning of each time step $t$, the decision maker (DM) has prediction $\widehat{\theta}_s^{(t)}$ about $\theta_s$ for $s \geq t$ and imperfect memory/information of past $\widehat{\theta}_s^{(t)}$ for $1 \leq s \leq t - 1$, and decides $x_t \in \mathcal{X}_t \subset \mathbb{R}^{d_t}$. Then he is given additional information about the true $\boldsymbol{\theta}^*$ (e.g. the exact value of $\theta_t^*$), and updates his information about the parameter sequence to $\widehat{\boldsymbol{\theta}}^{(t+1)} = (\widehat{\theta}_1^{(t+1)}, \widehat{\theta}_2^{(t+1)}, \ldots, \widehat{\theta}_T^{(t+1)})$.

The performance of the DM's output sequence $\mathbf{x} = (x_1, x_2, \ldots, x_T)$ is compared to the minimum of Eq. 1 over $\mathcal{X} := \prod_{t=1}^{T} \mathcal{X}_t$, and is evaluated using the dynamic regret defined as[1]

$$C(\mathbf{x}; \boldsymbol{\theta}^*) - C(\mathbf{x}^*; \boldsymbol{\theta}^*), \quad \mathbf{x}^* \in \operatorname*{arg\,min}_{\mathbf{x}' \in \mathcal{X}} C(\mathbf{x}'; \boldsymbol{\theta}^*).$$

*Motivating example 1: aggregate information.* Positive $(a, b)$ can model objectives that depend on "aggregate information" of the decision sequence, such as higher order finite differences and moving averages ($\frac{1}{a+b+1} \sum_{s=t-a}^{t+b} x_s$). This generalizes Li et al. (2021); Li & Li (2020): $(a, b) = (1, 0)$, $f_t(x_{t-1}, x_t; \theta_t^*) = \tilde{f}_t(x_t; \theta_t^*) + d_t(x_t, x_{t-1})$, with stage cost $\tilde{f}_t$ and switching cost $d_t$.

*Motivating example 2: decision making in advance, or delayed decision making.* If $f_t$ depends only on $x_{t-a}$ ($x_{t+b}$) for all $t$, then the decision made at time $s$, $x_s$, affects $f_{s+a}$ ($f_{s-b}$), i.e. the decision is made $a$-step in advance ($b$-step delayed). The window $[t - a, t + b]$ allows a combination of in-advance decisions up to $a$ steps and delayed decisions up to $b$ steps.

*Motivating example 3: parameters as dual variables.* For convex optimization with constraints which satisfy strong duality, one might aim at solving the Lagrangian relaxation, where the dual variables can be interpreted as (part of the) parameters. One might have predictions for the dual variables based on prior information or past data about the model. See Section A.1.

Following Li & Li (2020), we consider the setting where, in addition to predictions of the future parameters, the DM has access to a feasible point $\mathbf{x}^{(init)} \in \mathcal{X}$ in an online-with-look-ahead manner.

**Definition 1.1** ($k$-step look ahead initialization)**.** *We say that the DM has a $k$-step look ahead initialization, if there exists a feasible point $\mathbf{x}^{(init)} \in \mathcal{X}$, and the DM, at time $t$, has access to $x_s^{(init)}$ for $s = 1, 2, \ldots, \min(T, t + k)$.*

In general, any $\mathbf{x} \in \mathcal{X}$ can be used as a $k$-step look ahead initialization for any $k \in [T]$. However, as will be seen, the regret of the output of our algorithms depends on how good $\mathbf{x}^{(init)}$, in terms of $\|\mathbf{x}^{(init)} - \mathbf{x}^*\|$. To find a good initialization, one might take advantage of existing online convex optimization algorithms such as online gradient descent or online mirror descent (Li & Li, 2020).

---

[1]By Assumption 1.2 below, $\mathcal{X}$ is convex and compact, and $C(\cdot; \boldsymbol{\theta}^*)$ is continuous. Thus there exists $\mathbf{x}^* \in \mathcal{X}$ achieving the minimum.

## 1.2 MAIN RESULTS

On a high level, the predictions of $\boldsymbol{\theta}^*$ and the $k$-step look ahead initialization allow the DM to perform the classical (accelerated) gradient descent for $k/(a+b)$ steps to $\mathbf{x}^{(init)}$ using $\nabla C(\cdot; \widehat{\boldsymbol{\theta}})$, where $\widehat{\boldsymbol{\theta}}$ is chosen based on $\{\widehat{\boldsymbol{\theta}}^{(1)}, \ldots, \widehat{\boldsymbol{\theta}}^{(T)}\}$ to ensure *online-implementability*. Then, the regret and its dependency on parameter prediction errors follow naturally from properties of these *classical offline algorithms* — convergence rate and robustness against gradient inaccuracy, respectively.

To quantify how good the information $\widehat{\boldsymbol{\theta}}^{(t)}$ is at time $t$, we assume that $\Theta$ is a normed space, and that $\|\widehat{\theta}_s^{(t)} - \theta_s^*\|$ measures the error in prediction ($s \geq t$) or imperfect memory ($1 \leq s \leq t-1$). Further, we assume that $\nabla f_t$ is Lipschitz w.r.t. $\theta_t$. [2]

**Assumption 1.1** ($\nabla f_t$ is Lipschitz w.r.t. $\theta_t$).

$$|\frac{\partial f_t}{\partial x_s}(x_{W_t}; \theta_t) - \frac{\partial f_t}{\partial x_s}(x_{W_t}; \theta_t')| \leq h_{t,s}\|\theta_t - \theta_t'\|, \quad \forall x_{W_t} \in \mathcal{X}_{W_t}, \ s \in W_t, \ \theta_t, \theta_t' \in \Theta. \quad (2)$$

In addition, we make the following assumptions on the convexity of the objective function $C(\cdot; \boldsymbol{\theta}^*)$.

**Assumption 1.2** (smooth, (strongly) convex $C$ w.r.t. $\mathbf{x}$). $\mathcal{X} = \prod_{t=1}^T \mathcal{X}_t$ *where each $\mathcal{X}_t$ is compact and convex. For any $\boldsymbol{\theta} \in \Theta^T$, $C(\cdot; \boldsymbol{\theta}) : \mathcal{X} \to \mathbb{R}$ is convex and differentiable on $\mathcal{X}$. In addition, there exists $\kappa \in [0, 1]$, such that*

$$\frac{\kappa}{2}\|\mathbf{x}-\mathbf{y}\|^2 \leq C(\mathbf{x}; \boldsymbol{\theta}) - C(\mathbf{y}; \boldsymbol{\theta}) - \langle \nabla C(\mathbf{y}; \boldsymbol{\theta}), \mathbf{x}-\mathbf{y} \rangle \leq \frac{1}{2}\|\mathbf{x}-\mathbf{y}\|^2, \quad \forall \mathbf{x}, \mathbf{y} \in \mathcal{X}, \forall \boldsymbol{\theta} \in \Theta^T. \quad (3)$$

In terminology of convex optimization, $C(\cdot; \boldsymbol{\theta})$ is 1-smooth, and when $\kappa > 0$, it's also $\kappa$-strongly convex. Li & Li (2020); Li et al. (2021) assume that $\kappa > 0$, thereby their results hold only for strongly convex $C$. As will be seen, our algorithms provide guarantees even in the case when $\kappa = 0$, i.e. when the objective is not necessarily strongly convex. Moreover, Assumption 1.2 is weaker than assumptions on each $f_t$, and typically, one can think of $\kappa = \Theta(1)$ as a constant that does not depend on $T$ (see Section A.2).

Due to the constraint $\mathcal{X}_t$, we make the following assumption common in convex optimization literature:

**Assumption 1.3** (efficient projection). *For all $t \in [T]$, for all $y \in \mathbb{R}^{d_t}$, projecting $y$ to $\mathcal{X}_t$, i.e. finding $\arg\min_{x_t \in \mathcal{X}_t} \|x_t - y\|^2$, can be computed efficiently.*

Below, we state the performance of our algorithms online-PGM and online-AGM, which is to be presented in Algorithm 3.

**Theorem 1.1.** *Under the Assumptions 1.1, 1.2, 1.3, suppose that the DM has access to a $k$-step look ahead initialization as defined in 1.1, and that $L$ is chosen such that $(a + b)L \leq k$, that $\widehat{\boldsymbol{\theta}}^{(t)}$ is available at time $t$, and that $\kappa$ is given. Then Algorithm 3 outputs $\overline{x}_t$ at time $t = 1, 2, \ldots, T$ such that $\overline{\mathbf{x}}$ satisfies the following properties:*

• *For $\kappa = 0$,*

$$C(\overline{\mathbf{x}}; \boldsymbol{\theta}^*) - C(\mathbf{x}^*; \boldsymbol{\theta}^*) \leq \begin{cases} \frac{1}{2L}\mathcal{R}_0^2 + \frac{2D_{\mathcal{X}}}{L}\sum_{l=1}^L \epsilon_l & \text{(online-PGM)} \\ \frac{4}{L^2}\mathcal{R}_0^2 + \frac{4D_{\mathcal{X}}}{L^2}\sum_{l=1}^L l^2\epsilon_l & \text{(online-AGM).} \end{cases}$$

• *For $\kappa > 0$,*

$$C(\overline{\mathbf{x}}; \boldsymbol{\theta}^*) - C(\mathbf{x}^*; \boldsymbol{\theta}^*) \leq \begin{cases} \frac{\exp(-\kappa L/4)}{2}\mathcal{R}_0^2 + \frac{\kappa/8+1/4}{1-\rho^L}\sum_{l=1}^L \rho^{L-l}\epsilon_l^2 & \text{(online-PGM)} \\ 6\rho_1^L\mathcal{R}_0^2 + (1/\kappa + 1/2)\sum_{l=1}^L \rho_1^{L-l}\epsilon_l^2 & \text{(online-AGM),} \end{cases}$$

*where $\rho = 1 - \kappa/4$, $\rho_1 = (1 + \frac{1}{4}\sqrt{\kappa})^{-2}$, $\mathcal{R}_0 := \|\mathbf{x}^{(init)} - \mathbf{x}^*\|$, $D_{\mathcal{X}} = \max_{\mathbf{x}, \mathbf{x}' \in \mathcal{X}}\|\mathbf{x} - \mathbf{x}'\|$ is the diameter of $\mathcal{X}$. For $t \in [T]$, denoting $\widetilde{W}_t := [\max(1, t-b), \min(t+a, T)]$*

$$\epsilon_l^2 := \sum_{t=1}^{(a+b)(L-l)} (\sum_{s \in \widetilde{W}_t} h_{s,t}\|\widehat{\theta}_s^{(1)} - \theta_s^*\|)^2 + \sum_{t=(a+b)(L-l)+1}^{T} (\sum_{s \in \widetilde{W}_t} h_{s,t}\|\widehat{\theta}_s^{(t-(a+b)(L-l))} - \theta_s^*\|)^2.$$

---

[2]Alternatively, one can replace the norm on the RHS of Eq. 2 with a penalty function $\rho : \Theta \times \Theta \to \mathbb{R}_+$. Then our main results still hold (with potentially different constant factors), with $\|\theta_t - \widehat{\theta}_t\|$ replaced by $\rho(\theta_t, \widehat{\theta}_t)$.

When $\kappa > 0$, we can further use the upper bound that $\frac{\kappa}{2} \mathcal{R}_0^2 \leq C(\mathbf{x}^{(init)}; \boldsymbol{\theta}^*) - C(\mathbf{x}^*; \boldsymbol{\theta}^*)$, which implies that for online-AGM, there exists $\rho_1 = 1 - O(\sqrt{\kappa})$,

$$C(\overline{\mathbf{x}}; \boldsymbol{\theta}^*) - C(\mathbf{x}^*; \boldsymbol{\theta}^*) = O\left(\kappa^{-1} \rho_1^L (C(\mathbf{x}^{(init)}; \boldsymbol{\theta}^*) - C(\mathbf{x}^*; \boldsymbol{\theta}^*))\right) + O\left((\kappa^{-1} + 1) \sum_{l=1}^{L} \rho_1^{L-l} \epsilon_l^2\right).$$

### 1.3 CONTRIBUTIONS

Li & Li (2020) shows that it's possible to reduce the dynamic regret of $\mathbf{x}^{(init)}$ by a factor of $O(\kappa^{-1} \rho^k)$ for $\rho = 1 - \frac{\kappa}{4}$, at the cost of an additive term $O((\kappa^{-1} + 1) \sum_{l=1}^{k} \rho^{l-1} \delta_l))$ that depends on the $l$-step prediction errors $\delta_l$. A lower bound $\Omega(C_\kappa \sum_{t=1}^{T} \rho_0^{t-1} \delta_t)$ for $\rho_0 = (\frac{1-\sqrt{\kappa}}{1+\sqrt{\kappa}})^2$ is also proposed where $C_\kappa$ is a constant depending on $\kappa$. When $k$-step *exact prediction* is available, RHAG proposed in Li et al. (2021) uses accelerated gradient descent and can reduce the regret of the initialization by a factor of $O(\kappa^{-1} \rho_0^k)$. For the setup studied in Li & Li (2020); Li et al. (2021), our online-PGM is a slight variation to RHIG and achieves similar performance as RHIG, while our online-AGM is a slight variation to RHAG, and our results hold for the case when the gradients are inexact.

**Our contributions.** We show that acceleration is also possible when the prediction is inexact, and closes the gap on the decay rate ($\rho$ and $\rho_0$) of the influence of long-term prediction error. We propose an algorithm, online Accelerated Gradient Method, which performs accelerated gradient descent steps instead of gradient descent steps as in Li & Li (2020). Our online-AGM constructs solutions whose dynamic regret is the sum of two components: one term — $O(\kappa^{-1} \rho_1^{\frac{k}{a+b}} (C(\mathbf{x}^{(init)}; \boldsymbol{\theta}^*) - C(\mathbf{x}^*; \boldsymbol{\theta}^*)))$ — depends on how good the initialization is, and the other term — $O((\kappa^{-1} + 1) \sum_{l=1}^{\frac{k}{a+b}} \rho_1^{l-1} \delta_l))$ — depends on the prediction error. Importantly, $\rho_1 = 1 - O(\sqrt{\kappa})$, which depends on $\sqrt{\kappa}$ as in the lower bound rate $\rho_0$, and is smaller than the rate $\rho = 1 - \kappa/4$ for RHIG (for small enough $\kappa$).

In addition, we analyze the performance of online-PGM and online-AGM when the strong-convexity assumption is relaxed — a setting not studied in Li et al. (2021); Li & Li (2020) — and show that the regret decays at the rate of $O((\frac{k}{a+b})^{-1})$ and $O((\frac{k}{a+b})^{-2})$ respectively, with additive error terms due to the prediction inaccuracy. To the best of our knowledge, our online-PGM and online-AGM are the first *gradient-based* algorithms for smoothed online convex optimization (and objectives with more general couplings) with inexact predictions without the strong-convexity assumption.

As a by-product, we formalize and generalize Li & Li (2020) and Li et al. (2021)'s "fill-the-table" approach to running (accelerated) gradient descent online. We view the iterative updates in offline algorithms as state-evolution (in networks), and provide a general recipe to turn offline algorithms to online ones while maintaining the offline performance (such as convergence rate and robustness). This systematic approach to constructing online algorithms from offline ones might have applications in other problems.

### 1.4 NOTATIONS

We use boldface to denote variables that have $T$ components, such as $\mathbf{x} = (x_1, x_2, \ldots, x_T)$ and $\boldsymbol{\theta} = (\theta_1, \theta_2, \ldots, \theta_T)$. For convenience, for any $A \subset [T]$, we use $x_A$ to denote $(x_t)_{t \in A}$, and similarly for $\boldsymbol{\theta}$. We let $[n] := \{1, 2, \ldots, n\}$ for all $n \in \mathbb{N}$. For any convex compact set $\mathcal{K} \subset \mathbb{R}^d$ and $y \in \mathbb{R}^d$, $\mathsf{Proj}_{\mathcal{K}}(y) := \arg\min_{y' \in \mathcal{K}} \|y' - y\|^2$.

## 2 CONNECTIONS WITH PREVIOUS WORKS

**Smooth online convex optimization.** Our problem is motivated by a recent line of work on online convex optimization with switching cost, where the goal is to minimize $\sum_{t=1}^{T} f_t(x_t) + d(x_t, x_{t-1})$ by choosing $x_t$ sequentially, based on past decisions and past $f_t$'s, together with potentially inexact predictions about future $f_t$'s(Kwon & Pearson, 1977; Lin et al., 2012; Chen et al., 2016; Li et al., 2021; Li & Li, 2020). Various methods have been proposed to take advantage of the prediction. To name a few, RHC (Kwon & Pearson, 1977), AFHC (Lin et al., 2012) and CHC (Chen et al., 2016) choose $x_t$'s based on the *optimal solution* to the *predicted problem* restricted to windows around $t$; RHAG (Li et al., 2021) applies accelerated gradient descent with exact prediction, and RHIG (Li & Li, 2020) applies gradient descent with inexact prediction (more comparison in Section 1.3).

**Online convex optimization and dynamic regret.** We measure the performance of $\mathbf{x}$ using the dynamic regret, i.e. against the optimal $\mathbf{x}^*$ which does not necessarily satisfy $x_t^* = x_{t+1}^*$ for all $t$. Dynamic regret has been well studied for problems where $f_t$ depends only on $x_t$ (see Zinkevich (2003); Besbes et al. (2015); Zhao & Zhang (2021); Hazan (2022) and references therein). Typically, the regret is upper bounded using a combination of $T$, $\mathcal{P}_T$ the variation of the sequence $(x_1^*, \ldots, x_T^*)$, and/or $\mathcal{V}_T$ the variation of the sequence $(f_1, \ldots, f_T)$. For instance, the online gradient descent (OGD) achieves $O(\sqrt{T\mathcal{P}_T})$ (Zinkevich, 2003), and the restarted OGD achieves $O(T^{2/3}\mathcal{V}_T^{1/3})$ (Besbes et al., 2015). However, it's non-trivial to obtain dynamic regret guarantees for the general coupled objective functions where $f_t$ also depends on decisions made in the past and/or future. Li & Li (2020) shows that in the special setting of smooth online convex optimization with prediction, where the coupling is only due to the switching costs between consecutive decisions, restarted OGD can achieve $O(\sqrt{T\mathcal{V}_T})$ dynamic regret, with additive terms due to prediction errors.

**Other related online optimization problems.** Convex optimization with memory (Anava et al., 2015; Kumar et al., 2023; Shi et al., 2020) can be viewed as a special case of our problem Eq. 1 with $b = 0$. However, static regret and the offline fixed decision are usually used as benchmarks. Also related is online optimization with prediction, where bound on static regret using the prediction error has been obtained for online mirror descent (Rakhlin & Sridharan, 2013).

**Smooth convex optimization with inexact oracles.** Under Assumption 1.1, prediction error can be related to error in gradient, and thus be treated as a form of oracle inaccuracy. Our Algorithm 3 builds upon Devolder et al. (2013a) and Devolder et al. (2013b), which study the convergence properties of (accelerated) gradient descent with inexact first order oracle. We present a modification of their results below in Section 3.1. Optimization with inexact oracle has also been studied in many other works: d'Aspremont (2008) and Schmidt et al. (2011), to name a few.

**Decentralized convex optimization.** Our objective function $C(x_1, x_2, \ldots, x_T; \boldsymbol{\theta})$ can be viewed as a function of $T$ components and fits naturally into a network model, where each vertex represents the decision at some time step, and vertices communicate information such as current decision variables, gradients, and momentum. This connects our problem with many other network-related problems, especially parallel/distributed optimization (Scaman et al., 2017; Mosk-Aoyama et al., 2010; Bertsekas & Tsitsiklis, 2015). It will be interesting to further explore what insights these network-related problems can bring to our online convex optimization with prediction.

# 3 TWO INGREDIENTS IN ALGORITHM DESIGN

Our online-PGM and online-AGM (Algorithm 3) can be viewed as *offline convex optimization algorithms which are robust to oracle errors*, implemented in an *asynchronous* fashion such that the updates can be carried out online. We explain these two ingredients in Sections 3.1 and 3.2.

## 3.1 OPTIMIZATION WITH INEXACT FIRST ORDER ORACLE

Offline smooth convex optimization with first order oracle is a well studied problem, and the accelerated gradient method is known to achieve the optimal convergence rates of $O(1/k^2)$ and of $O(\exp(-\sqrt{\kappa}k))$ for strongly convex $\kappa$-conditioned objectives(Bubeck, 2015). In fact, Devolder et al. (2013a) and Devolder et al. (2013b) show that the accelerated gradient method is also robust to gradient inaccuracy: the convergence rates of $O(1/k^2)$ and $\exp(-\sqrt{\kappa}t)$ still hold, but the suboptimality gaps have one extra additive term that depends on the error in the gradients. This applies exactly to our setting, where $\nabla C(\cdot; \widehat{\boldsymbol{\theta}})$ is used as an approximation to $\nabla C(\cdot; \boldsymbol{\theta}^*)$.

Formally, Devolder et al. (2013a) and Devolder et al. (2013b) study convex optimization with inexact first order oracle. The goal is to solve the following convex optimization problem

$$\min_{x \in \mathcal{K}} F(x) \tag{4}$$

where $\mathcal{K} \subset \mathbb{R}^d$ is closed and convex, and $F$ is convex on $\mathcal{K}$, and the optimal is achieved at some $x^* \in \mathcal{K}$. The algorithm has access to a $(\delta, m, M)$ oracle defined as

**Definition 3.1** (($\delta, m, M$)-oracle). *We say $\mathcal{O} : \mathcal{K} \to \mathbb{R} \times \mathbb{R}^d$ is a first-order $(\delta, m, M)$-oracle if for any $\mathbf{y} \in \mathcal{K}$, when queried at $y$, the oracle returns $(F_{\delta,m,M}(\mathbf{y}), g_{\delta,m,M}(\mathbf{y})) \in \mathbb{R} \times \mathbb{R}^d$ such that*

$$\frac{m}{2}\|x - y\|^2 \leq F(x) - F_{\delta,m,M}(y) - \langle g_{\delta,m,M}(\mathbf{y}), x - y \rangle \leq \frac{M}{2}\|x - y\|^2 + \delta, \quad \forall x \in \mathcal{K}, \quad (5)$$

*where $\delta \geq 0$, $0 \leq m \leq M$.*

The simplest example is $\mathcal{O}(x) = (F(x), \nabla F(x))$, which is a $(0, m, M)$-oracle when $F$ is $m$-strongly convex and $M$-smooth. In fact, Devolder et al. (2013a) and Devolder et al. (2013b) show that if one has an inexact gradient and function value oracle for $F$, one can construct a $(\delta, m', M')$-oracle for some $\delta, m', M'$ that might depend on $m, M$ and error in gradient and value oracles (Proposition B.1).

| Method | Assumption | Evaluation $\overline{x}^{(l)}$ | Dependency on $\mathcal{R}_0$ | Dependency on $\delta_l$'s |
|---|---|---|---|---|
| PGM | $m > 0$ | $\frac{\kappa \sum_{i=1}^{l}(1-\kappa)^{l-i}x^{(i)}}{1-(1-\kappa)^l}$ | $\frac{M}{2}\exp(-\kappa l)\mathcal{R}_0^2$ | $\frac{\kappa \sum_{i=1}^{l}(1-\kappa)^{l-i}\delta_i}{1-(1-\kappa)^l}$ |
| PGM | $m = 0$ | $\frac{1}{l}\sum_{i=1}^{l}x^{(i)}$ | $\frac{M}{2l}\mathcal{R}_0^2$ | $\frac{1}{l}\sum_{i=1}^{l}\delta_i$ |
| AGM | $m > 0$ | $x^{(l)}$ | $\left(1 + \frac{\sqrt{\kappa}}{2}\right)^{-2l} \cdot 3M\mathcal{R}_0^2$ | $\sum_{i=1}^{l}\left(1 + \frac{\sqrt{\kappa}}{2}\right)^{-2(l-i)}\delta_i$ |
| AGM | $m = 0$ | $x^{(l)}$ | $\frac{4M}{l^2}\mathcal{R}_0^2$ | $4\sum_{i=1}^{l}\left(\frac{i}{l}\right)^2 \delta_i$ |

Table 1: Convergence properties for convex optimization with $(\delta_l, m, M)$-oracle $\mathcal{O}^{(l)}$, $l = 1, 2, \ldots$. Denote $\mathcal{R}_0 := \|x^{(0)} - x^*\|$ and $\kappa = m/M$, and the guarantee is $F(\overline{x}^{(l)}) - F(x^*) \leq$ "Dependency on $\mathcal{R}_0$" + "Dependency on $\delta_l$'s". See Theorem B.1 and B.2 for the exact statements.

In Table 1, we summarize the performance of the Projected Gradient Method (PGM, Algorithm 1), and the Accelerated Gradient Method (AGM, Algorithm 2) proposed in Devolder et al. (2013a;b) for problems with inexact first order oracles. The proofs for Theorem B.1 and B.2, adapted from Devolder et al. (2013a) to deal with iteration-dependent $\delta_l$ (at iteration $l$, the oracle $O^{(l)}$ is a $(\delta_l, m, M)$-oracle), are provided in Appendix B.

---

**Algorithm 1:** Projected Gradient Method with $(\delta, m, M)$-oracle

**Input:** Initial $x^{(0)} \in \mathcal{K}$, $\mathcal{O}^{(l)}$ an $(\delta_l, m, M)$-oracle for $F$ for $l = 1, 2, \ldots$
**Output:** $x^{(1)}, x^{(2)}, \ldots$
**for** $l = 1, 2, \ldots,$ **do**
    Obtain $(f^{(l-1)}, g^{(l-1)}) \leftarrow \mathcal{O}^{(l)}(x^{(l-1)})$;
    Update $x^{(l)} \leftarrow$
    $\arg\min_{x \in \mathcal{K}} \langle g^{(l-1)}, x - x^{(l-1)} \rangle + \frac{M}{2}\|x - x^{(l-1)}\|^2$;
**end**

---

**Algorithm 2:** Accelerated Gradient Method with $(\delta, m, M)$-oracle

**Input:** Initial $x^{(init)} \in \mathcal{K}$, $\mathcal{O}^{(l)}$ an $(\delta_l, m, M)$-oracle for $F$ for $l = 1, 2, \ldots$, sequence $(\alpha_l)_{l \in \mathbb{N}}$ and sequence $(\tau_l)_{l \in \mathbb{N}}$.
**Output:** the sequence $x^{(1)}, x^{(2)}, x^{(3)}, \ldots$
Initialize $y^{(1)} \leftarrow x^{(init)}$, $v^{(0)} \leftarrow 0 \in \mathbb{R}^d$;
**for** $l = 1, 2, \ldots,$ **do**
    Obtain $(f^{(l)}, g^{(l)}) \leftarrow \mathcal{O}^{(l)}(y^{(l)})$;
    Compute

$$x^{(l)} \leftarrow \arg\min_{x \in \mathcal{K}} \langle g^{(l)}, x - y^{(l)} \rangle + \frac{M}{2}\|x - y^{(l)}\|^2$$

$$v^{(l)} \leftarrow v^{(l-1)} + \alpha_l(g^{(l)} - my^{(l)})$$

$$z^{(l)} \leftarrow \arg\min_{y \in \mathcal{K}} \frac{M}{2}\|y - y^{(1)}\|^2 + \langle v^{(l)}, y \rangle + \frac{m(\sum_{j=1}^{l}\alpha_j)}{2}\|y\|^2$$

    Update $y^{(l+1)} \leftarrow \tau_l z^{(l)} + (1 - \tau_l)x^{(l)}$;
**end**

---

### 3.2 FROM OFFLINE ALGORITHMS TO ONLINE ALGORITHMS

The second observation is that offline algorithms that update all variables synchronously can be implemented in an asynchronously manner, such that the variables can be updated sequentially. In fact, this can be done efficiently as long as the variables are only "weakly coupled". Take PGM as an example. When applying PGM to our objective $C(\cdot; \boldsymbol{\theta}^*)$, in the $l$-th iteration, the update is $\mathbf{x}^{(l)} = \text{Proj}_{\mathcal{X}}(\mathbf{x}^{(l-1)} - \eta \nabla C(\mathbf{x}^{(l-1)}; \boldsymbol{\theta}^*))$ for some $\eta \geq 0$. Direct computation shows that

$$\frac{\partial C}{\partial x_t}(\mathbf{x}; \boldsymbol{\theta}^*) = \sum_{s=\max(1,t-b)}^{\min(t+a,T)} \frac{\partial f_s}{\partial x_t}(x_{W_s}; \theta_s^*), \quad \forall \mathbf{x} \in \mathcal{X} \quad (6)$$

In particular, this implies that $\frac{\partial C}{\partial x_t}(\mathbf{x}; \boldsymbol{\theta}^*)$ depends only on $x_{s'}$ for $s' \in \cup_{s=\max(1,t-b)}^{\min(t+a,T)} W_s$, that is, for $|s' - s| \leq a + b$. See Figure 1 for an example for $a = 2, b = 1$.

Figure 1: Example: $a = 2, b = 1$, $\frac{\partial C}{\partial x_t}(\mathbf{x}; \boldsymbol{\theta}) = \sum_{s=t-1}^{t+2} \frac{\partial f_s}{\partial x_t}(x_{(s-2):(s+1)}; \theta_s)$ depends on $\theta_{(t-1):(t+2)}$ and $x_{(t-3):(t+3)}$.

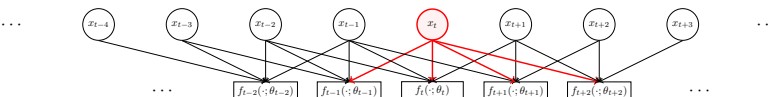

In other words, $x_t^{(l)}$ can be computed as long has $x_s^{(l-1)}$ has been computed for all $|s - t| \le a + b$, even if some $x_{s'}^{(l-1)}$ has not been computed. This idea is used in the design of RHIG algorithm, where $a = 1, b = 0$ and simple gradient descent steps like those in Algorithm 1 are used.

Below, in Section 3.2.1 we first look at a hypothetical offline setting for the problem Eq. 1, which can be solved using Algorithms 1 (PGM) and 2 (AGM). In Section 3.2.2, we show that the iterative updates in PGM and AGM admit a state-transition view, where roughly the states can be taken as $x$ for PGM, and $(x, v)$ for AGM. In Section 3.2.3, we explore this state-transition view in a more general network setting, and propose a recipe to simulate synchronous state-evolution asynchronously, thereby turning offline algorithms into online ones. This "network and states" perspective allows us to work with objectives that have more complex dependency structure (e.g. $a, b \ge 1$) in a unified way (the "momentum term" $v$ in AGM is treated as part of the state).

### 3.2.1 A HYPOTHETICAL OFFLINE CONVEX OPTIMIZATION PROBLEM

First, recall that Equation 6 implies that $\frac{\partial C}{\partial x_t}(\mathbf{x}; \boldsymbol{\theta}) = \frac{\partial C}{\partial x_t}(x_{\overline{W}_t}; \theta_{\widetilde{W}_t})$ depends only on $x_{\overline{W}_t}$ where $\overline{W}_t := [\max(1, t-a-b), \min(t+a+b, T)]$, and on $\theta_{\widetilde{W}_t}$ where $\widetilde{W}_t := [\max(1, t-b), \min(t+a, T)]$. This allows us to define, for each $\theta_{\widetilde{W}_t} \in \Theta^{|\widetilde{W}_t|}$, $G_t(\cdot; \theta_{\widetilde{W}_t}) : \mathcal{X}_{\overline{W}_t} \to \mathbb{R}^{d_t}$ as

$$G_t(x_{\overline{W}_t}; \theta_{\widetilde{W}_t}) := \frac{\partial C}{\partial x_t}(x_{\overline{W}_t}; \theta_{\widetilde{W}_t}) = \sum_{s \in \widetilde{W}_t} \frac{\partial f_s}{\partial x_t}(x_{W_s}; \theta_s), \quad \forall x_{\overline{W}_t} \in \mathcal{X}_{\overline{W}_t}. \tag{7}$$

Next, we consider solving the problem Eq. 1, but in the following hypothetical offline convex optimization setting: the algorithm is given an initial feasible solution $\mathbf{x}^{(init)} \in \mathcal{X}$, and a sequence of oracles $\mathcal{O}^{(l)}$ for $l = 1, 2, \ldots, L$, such that

$$\mathcal{O}^{(l)}(\mathbf{x}) = (0, G_1(x_{\overline{W}_1}; \theta_{\widetilde{W}_1}^{(l)}), G_2(x_{\overline{W}_2}; \theta_{\widetilde{W}_2}^{(l)}), \ldots, G_T(x_{\overline{W}_T}; \theta_{\widetilde{W}_T}^{(l)})), \quad \forall \mathbf{x} \in \mathcal{X}, \tag{8}$$

where $\theta_{\overline{W}_t}^{(l)} \in \Theta^{|\overline{W}_t|}$ for all $l \in [L]$ and $t \in [n]$. $G_t$ can be viewed as an approximation to $\frac{\partial C}{\partial x_t}$. For instance, when $\boldsymbol{\theta} = \boldsymbol{\theta}^*$, $G_t(x_{\overline{W}_t}; \theta_{\widetilde{W}_t}^*) = \frac{\partial C}{\partial x_t}(x_{\overline{W}_t}; \theta_{\widetilde{W}_t}^*)$, and in general, the norm of gradient error is upper bounded by $(\sum_{t=1}^{T}(\sum_{s \in \widetilde{W}_t} h_{s,t}\|\theta_s^* - \theta_s^{(l)}\|)^2)^{1/2}$ by Assumption 1.1 (Proposition C.1).

Consider Algorithms 1 and 2 with the oracles $\mathcal{O}^{(1)}, \mathcal{O}^{(2)}, \ldots, \mathcal{O}^{(L)}$. Notice that in our design of $\mathcal{O}^{(l)}$, the (approximate) function value term is always set to 0. Nevertheless, in both algorithms the (approximate) function values are never used. As a result, when running these algorithms, the updates are the same as if the (approximate) function values are not 0 but set to the exact function values at the queried points. In particular, taking $\Delta_2^{(l)} = (\sum_{t=1}^{T}(\sum_{s \in \widetilde{W}_t} h_{s,t}\|\theta_s^* - \theta_s^{(l)}\|)^2)^{1/2}$, if $\kappa = 0$, $\mathcal{O}^{(l)}$ is equivalent to a $(\delta_l, 0, 1)$-oracle for $C(\cdot; \boldsymbol{\theta}^*)$, where $\delta_l = 2\Delta_2^{(l)} D_{\mathcal{X}}$. If $\kappa > 0$, for $l \in [L]$, $\mathcal{O}^{(l)}$ is equivalent to a $(\delta_l, \frac{\kappa}{2}, 2)$-oracle for $C(\cdot; \boldsymbol{\theta}^*)$, where $\delta_l = (\frac{1}{\kappa} + \frac{1}{2})(\Delta_2^{(l)})^2$. Applying Theorem B.1 or Theorem B.2, we get the convergence rate (stated in Corollary C.1 and Corollary C.2).

### 3.2.2 STATE EVOLUTION PERSPECTIVE OF ALGORITHMS 1 AND 2

In fact, since the feasible set $\mathcal{X} = \prod_{t=1}^{T} \mathcal{X}_i$ and $G_t(\cdot; \theta_{\widetilde{W}_t}) : \mathcal{X}_{\overline{W}_t} \to \mathbb{R}^{d_t}$ depends only on $x_s$ for $s \in \overline{W}_t$, the update steps in Algorithms 1 and 2 are separable, in the following sense:
- in the $l$-th iteration of Algorithms 1, for $t = 1, 2, \ldots, T$,

$$x_t^{(l)} \leftarrow \mathsf{Proj}_{\mathcal{X}_t}(x_t^{(l-1)} - \frac{1}{M} G_t(x_{\overline{W}_t}^{(l-1)}; \theta_{\widetilde{W}_t}^{(l)})). \tag{9}$$

That is, $x_t^{(l)}$ depends only on $x_{\overline{W}_t}^{(l-1)}$ (and $\theta_{\widetilde{W}_t}^{(l)}$). Here $M = 1$ if $\kappa = 0$ and $M = 2$ if $\kappa > 0$.

- in the $l$-th iteration of Algorithms 2, for $t = 1, 2, \ldots, T$,

$$x_t^{(l)} \leftarrow \mathsf{Proj}_{\mathcal{X}_t}(y_t^{(l)} - \frac{1}{M}G_t(y_{\overline{W}_t}^{(l)}; \theta_{\widetilde{W}_t}^{(l)})), \quad v_t^{(l)} \leftarrow v_t^{(l-1)} + \alpha_l(G_t(y_{\overline{W}_t}^{(l)}; \theta_{\widetilde{W}_t}^{(l)}) - my_t^{(l)})$$

$$z_t^{(l)} \leftarrow \mathsf{Proj}_{\mathcal{X}_t}(\frac{My_t^{(1)} - v_t^{(l)}}{m(\sum_{j=1}^{l}\alpha_j) + M}), \quad y_t^{(l+1)} \leftarrow \tau_l z_t^{(l)} + (1 - \tau_l)x_t^{(l)}$$

Here $m = 0, M = 1$ if $\kappa = 0$, and $m = \kappa/2, M = 2$ if $\kappa > 0$. Notice that $z_t^{(l)}$ and $y_t^{(l)}$ depend only on $(x_t^{(l)}, v_t^{(l)} - My_t^{(1)})$, while $(x_t^{(l)}, v_t^{(l)} - My_t^{(1)})$ depends only on $(x_{\overline{W}_t}^{(l-1)}, v_{\overline{W}_t}^{(l-1)} - My_{\overline{W}_t}^{(1)})$ (together with $l$ and $\theta_{\widetilde{W}_t}^{(l)}$), thus $(x_t^{(l)}, v_t^{(l)} - My_t^{(1)})$ can be used as state variables.

A more precise definition of the states is presented in Section C.3. Consequently, in Algorithm 1, as long as $x_{\overline{W}_t}^{(l-1)}$ is computed before $x_t^{(l)}$ is computed, the evolution of $\mathbf{x}^{(l)}$ remains the same as if all $x_t$'s are updated at the same time. Similarly for Algorithm 2.

Figure 2: Example: $a = 2, b = 1$, one step gradient descent for $x_t$ using $G_t$, which depends on $x_{(t-3):(t+3)}$. For simplicity, we omit the dependency on $\boldsymbol{\theta}^{(l)}$.

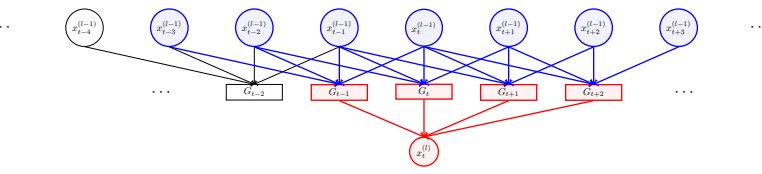

### 3.2.3 ASYNCHRONOUS UPDATE FOR SYNCHRONOUS ALGORITHMS

Let $G = (V, E)$ be an undirected graph. For each vertex $v \in V$ and $\xi = 0, 1, \ldots, \mathcal{N}_\xi(v) \subset V$ is the set of vertices that are $\xi$-hop away from $v$. For instance, $\mathcal{N}_0(v) = \{v\}$ and $\mathcal{N}_1(v)$ is the the neighbor of $v$, which we abbreviate as $\mathcal{N}(v)$. For convenience $\overline{\mathcal{N}_\xi}(v) = \cup_{i=0}^{\xi}\mathcal{N}_i(v)$, and $\overline{\mathcal{N}}(v) = \overline{\mathcal{N}_1}(v)$.

We associate $v \in V$ a state space $\mathcal{S}_v$ and a state update function $\phi_v : \mathcal{S}_{\overline{\mathcal{N}}(v)} \to \mathcal{S}_v$. Consider the following synchronous update: at $t = 0$, vertex $v$ is at state $s_v^{(0)} \in \mathcal{S}_v$. For $l = 1, 2, \ldots$, $s_v^{(l)} = \phi_v(s_{\overline{\mathcal{N}}(v)}^{(l-1)})$. That is, the state of $v$ at $l$ depends only on the states of $v$ and its neighbors at iterations $l - 1$. Importantly, the update order for the vertices within the $l$-th iteration does not matter, since the new states depend only on the states in the previous step. However, the update order cannot be changed across iterations (e.g. update $s_v$ twice at iteration $l$ and fix it at iteration $l + 1$).

For any fixed ordering $\sigma : [|V|] \to V, L \in \mathbb{N}$, in Algorithm 5 we give an asynchronous way to update the states for $L$ steps such that the state evolution $\tilde{s}_v^{(l)}$ for all $v \in V, l \in [L]$ satisfies

- consistency: the state evolution is the same as the synchronous update, i.e. $\tilde{s}_v^{(l)} = s_v^{(l)}$ for all $v \in V, l \in [L]$

- minimum information: for all vertex $i \in V$, the sequence $\tilde{s}_{\sigma(i)}^{(1:L)} = (\tilde{s}_{\sigma(i)}^{(1)}, \ldots, \tilde{s}_{\sigma(i)}^{(L)})$ does not depend on $s_v^{(0)}$ for $v \notin \cup_{j=1}^{i}\overline{\mathcal{N}_L}(\sigma(j))$

The second condition implies that $s_{\sigma(1)}^{(1:L)}, s_{\sigma(2)}^{(1:L)}, \ldots, s_{\sigma(|V|)}^{(1:L)}$ can be computed *sequentially*, such that by the time $\tilde{s}_{\sigma(i)}^{(1:L)}$ is computed, only $s_{\overline{\mathcal{N}_L}(\sigma(1))}^{(0)}, s_{\overline{\mathcal{N}_L}(\sigma(2))}^{(0)}, \ldots, s_{\overline{\mathcal{N}_L}(\sigma(i))}^{(0)}$ are revealed (some information might be revealed multiple times). In addition, we point out that the first property, consistency, holds for all $l \in [L]$ not just for the last iterate $l = L$. This is crucial since for many algorithms, the final output depends not just on the last iterate variables, but also on the entire path. For instance, for Algorithms 1 and 2, the performance guarantee is stated for some weighted average of the intermediate decision variables.

The Algorithm 5 and its performance (Theorem C.1) are stated in Appendix C.2. To illustrate the idea, let's take $V = [|V|]$ and $\sigma(i) = i$ for all $i \in V$, then Algorithm 5 initializes $\tilde{s}_{\overline{\mathcal{N}_L}(1)}^{(0)}$ using $s_{\overline{\mathcal{N}_L}(1)}^{(0)}$, then computes sequentially $\tilde{s}_{\overline{\mathcal{N}_{L-1}}(1)}^{(1)}, \tilde{s}_{\overline{\mathcal{N}_{L-2}}(1)}^{(2)}, \ldots, \tilde{s}_{\overline{\mathcal{N}_1}(1)}^{(L-1)}, \tilde{s}_1^{(L)}$. Since $\tilde{s}_{\overline{\mathcal{N}_{L-l}}(1)}^{(l)}$ depends only on $\tilde{s}_{\overline{\mathcal{N}_{L-l+1}}(1)}^{(l-1)}$, which has already been computed when $\tilde{s}_{\overline{\mathcal{N}_{L-l}}(1)}^{(l)}$ is computed, this

order of update is valid. Then $\tilde{s}^{(0)}_{\overline{\mathcal{N}}_L(2)}$ is initialized using $s^{(0)}_{\overline{\mathcal{N}}_L(1)}$, and it computes sequentially $\tilde{s}^{(1)}_{\overline{\mathcal{N}}_{L-1}(2)}, \tilde{s}^{(2)}_{\overline{\mathcal{N}}_{L-2}(2)}, \ldots, \tilde{s}^{(L-1)}_{\overline{\mathcal{N}}_1(2)}, \tilde{s}^{(L)}_2$. Similarly for the rest of the nodes. (If $\tilde{s}^{(l)}_v$ has been set before, $\tilde{s}^{(l)}_v$ won't be computed again.) Figure 3 gives an example when $L = 2$.

Figure 3: Update rule for Algorithm 5 for $v = 1, 2$, $L = 2$.

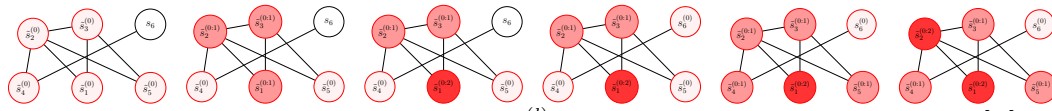

For Algorithms 1 and 2 run with the oracles $\mathcal{O}^{(l)}$ defined as in Eq. 8, we can take $V = [T]$ and $E = \{\{u, v\} \subset V^2, \ u \neq v, \ |u - v| \leq a + b\}$. Then the neighborhood of $t \in V$ is $\overline{\mathcal{N}}(t) = \overline{W}_t$, with the states as defined in Section C.3.

## 4 ALGORITHMS AND PERFORMANCE

Finally, we present our Algorithm 3: online-PGM in green , and online-AGM in yellow . Just like the offline PGM (Algorithm 1), for online-PGM, each update is a projected gradient descent update using $G_{s,l} = G_s(x^{(l-1)}_{\overline{W}_s}; \theta^{(l)}_{\overline{W}_s}) \approx \frac{\partial C}{\partial x_s}(\mathbf{x}^{(l-1)}; \boldsymbol{\theta}^*)$ ($G_s$ is defined in Eq. 7)

$$x^{(l)}_s \leftarrow \mathsf{Proj}_{\mathcal{X}_s}(x^{(l-1)}_s - \frac{1}{M}G_{s,l}). \tag{10}$$

Similarly, following the offline AGM (Algorithm 2), in online-AGM, each update is a projected gradient descent step at the extrapolated point $\mathbf{y}^{(l)}$, followed by an update on momentum terms and next query point. With $G_{s,l} = G_s(y^{(l)}_{\overline{W}_s}; \theta^{(l)}_{\overline{W}_s}) \approx \frac{\partial C}{\partial x_s}(\mathbf{y}^{(l)}; \boldsymbol{\theta}^*)$

$$x^{(l)}_s \leftarrow \mathsf{Proj}_{\mathcal{X}_s}(y^{(l)}_s - \frac{1}{M}G_{s,l}), \quad \tilde{v}^{(l)}_s \leftarrow \tilde{v}^{(l-1)}_s + \alpha_l(G_{s,l} - my^{(l)}_s), \quad y^{(l+1)}_s \leftarrow \tau_l z^{(l)}_s + (1-\tau_l)x^{(l)}_s, \tag{11}$$

where $z^{(l)}_s \leftarrow \mathsf{Proj}_{\mathcal{X}_s}\left(\frac{-\tilde{v}^{(l)}_s}{m(\sum_{j=1}^{l}\alpha_j)+M}\right)$.

Thus, it remains to choose $L$ and $\theta^{(l)}_{\overline{W}_v}$ for each iteration $l$ so that the asynchronous update rule in Algorithm 5 (with $\sigma(i) = i$ for $i \in [|V|]$) can be implemented online given $k$-step initialization ahead. It turns out that our update order can also be viewed as a "fill-the-table" type as in Li & Li (2020). Thus, $L \leq k/(a+b)$, and the following choice of $\theta^{(l)}_{\overline{W}_v}$'s are valid, with details in Appendix C.4:

$$\theta^{(l)}_{\overline{W}_v} = \widehat{\theta}^{(1)}_{\overline{W}_v}, \ v \in [(a+b)(L-l)+1], \quad \theta^{(l)}_{\overline{W}_v} = \widehat{\theta}^{(v-(a+b)(L-l))}_{\overline{W}_v}, \ v = (a+b)(L-l)+2, \ldots, T. \tag{12}$$

For online AGM, we make the further simplification: since a direct implementation of Algorithm 5 for AGM only keeps in the memory the state variables $((x, v))$ and recompute the non-state ones $((y, s))$. These repetitive computations can be inefficient, and when memory is not a concern, these dependent variables can be computed once, and stored. In Algorithm 3, we provide such simplified implementation.

*Proof of Theorem 1.1.* By Theorem C.1, the sequence $x^{(l)}_t$ generated when running Algorithm 3 is the same as the sequence when running Algorithm 1 and 2, with the inexact oracles $\mathcal{O}^{(l)}$ as in Eq. 8, and choice of $\boldsymbol{\theta}^{(l)}$ as in Eq. 12, the result follows from Corollary C.1 for the case $\kappa = 0$ and Corollary C.2 for the case $\kappa > 0$. □

## 5 NUMERICAL EXPERIMENTS

We compare the performance of online-PGM and online-AGM[3] using a variant of the planning problem in Li & Li (2020), stated below:

$$C(\mathbf{x}; \boldsymbol{\theta}) := \frac{1}{2}\sum_{t=1}^{T}(a_t(x_t - \theta_t)^2 + \frac{1}{2}(x_t - x_{t-1})^2), \quad \mathbf{x} \in \mathcal{X} := [-10^6, 10^6]^T. \tag{13}$$

---

[3]As pointed out in Section 1.3, online-PGM is a slight variation to RHIG and achieves similar theoretical performance. Thus, the comparisons in this experiment can be viewed as between RHIG and online-AGM.

**Algorithm 3:** online Projected Gradient Method and online Accelerated Gradient Method .

For $t \in [T]$, $\widetilde{W}_t := [\max(1, t-b), \min(t+a, T)]$, $\overline{W}_t := [\max(1, t-a-b), \min(t+a+b, T)]$

**Input:** $x_s^{(init)} \in \mathcal{X}_s$ for $s = 1, 2, \dots, T$, available for step $t \geq s - k$, $L (\leq \frac{k}{a+b})$ the number of updates, $\widehat{\theta}^{(t)} \in \Theta^T$ available at time $t$ for $t = 1, 2, \dots, T$. $\kappa \in [0, 1]$.

**Output:** $\overline{x}_1, \overline{x}_2, \dots, \overline{x}_T$

initialize $(m, M) \leftarrow (0, 1)$ if $\kappa = 0$, and $(m, M) \leftarrow (\frac{\kappa}{2}, 2)$ if $\kappa > 0$;

initialize $x_{1:(1+L(a+b))}^{(0)} \leftarrow x_{1:(1+L(a+b))}^{(init)}$;

initialize $y_{1:(1+L(a+b))}^{(1)} \leftarrow x_{1:(1+L(a+b))}^{(init)}$;

initialize $\tilde{v}_{1:(1+L(a+b))}^{(0)} \leftarrow -My_{1:(1+L(a+b))}^{(1)}$;

compute $(\alpha_l)_{l \in [L]}$ such that $\alpha_1 = 1$, and $\forall l \in [L]$

$(1 + \frac{m}{M}\sum_{i=1}^{l}\alpha_i)(\sum_{i=1}^{l+1}\alpha_i) = \alpha_{l+1}^2$;

compute $\tau_l := \frac{\alpha_{l+1}}{\sum_{j=1}^{l+1}\alpha_j}$;

**for** $l = 1, 2, \dots, L$ **do**
  **for** $s = 1, 2, \dots, 1 + (a+b)(L-l)$ **do**
    $G_{s,l} \leftarrow \sum_{s' \in \widetilde{W}_s} \frac{\partial f_{s'}}{\partial x_s}(x_{W_{s'}}^{(l-1)}; \widehat{\theta}_{s'}^{(1)})$;
    update $x_s^{(l)}$ using $G_{s,l}$ and Eq. 10;
    $G_{s,l} \leftarrow \sum_{s' \in \widetilde{W}_s} \frac{\partial f_{s'}}{\partial x_s}(y_{W_{s'}}^{(l)}; \widehat{\theta}_{s'}^{(1)})$;
    update $x_s^{(l)}, \tilde{v}_s^{(l)}, z_s^{(l)}, y_s^{(l+1)}$ using $G_{s,l}$ and Eq. 11;
  **end**
**end**

$\overline{x}_1 \leftarrow \begin{cases} \frac{1}{L}\sum_{l=1}^{L} x_1^{(l)} & (\kappa = 0) \\ \frac{m/M}{1-(1-\frac{m}{M})^L}\sum_{l=1}^{L}(1 - m/M)^{L-l}x_1^{(l)} & (\kappa > 0) \end{cases}$

$\overline{x}_1 \leftarrow x_1^{(L)}$;

**for** $t = 2, \dots, T$ **do**
  **if** $t + L(a+b) \leq T$ **then**
    initialize $x_{t+L(a+b)}^{(0)} \leftarrow x_{t+L(a+b)}^{(init)}$;
    initialize $y_{t+L(a+b)}^{(1)} \leftarrow x_{t+L(a+b)}^{(init)}$;
    initialize $v_{t+L(a+b)}^{(0)} \leftarrow -My_{t+L(a+b)}^{(1)}$;
  **end**
  **for** $l = 1, 2, \dots, L$ **do**
    **if** $t + (a+b)(L-l) \leq T$ **then**
      $s \leftarrow t + (a+b)(L-l)$;
      $G_{s,l} \leftarrow \sum_{s' \in \widetilde{W}_s} \frac{\partial f_{s'}}{\partial x_s}(x_{W_{s'}}^{(l-1)}; \widehat{\theta}_{s'}^{(t)})$;
      update $x_s^{(l)}$ using $G_{s,l}$ and Eq. 10;
      $G_{s,l} \leftarrow \sum_{s' \in \widetilde{W}_s} \frac{\partial f_{s'}}{\partial x_s}(y_{W_{s'}}^{(l)}; \widehat{\theta}_{s'}^{(t)})$;
      update $x_s^{(l)}, \tilde{v}_s^{(l)}, z_s^{(l)}, y_s^{(l+1)}$ using $G_{s,l}$ and Eq. 11;
    **end**
  **end**
  $\overline{x}_t \leftarrow \begin{cases} \frac{1}{L}\sum_{l=1}^{L} x_t^{(l)} & (\kappa = 0) \\ \frac{m/M}{1-(1-\frac{m}{M})^L}\sum_{l=1}^{L}(1 - m/M)^{L-l}x_t^{(l)} & (\kappa > 0) \end{cases}$
  $\overline{x}_t \leftarrow x_t^{(L)}$;
**end**

Here the parameter $\theta$ is composed of a known sinusoidal signal term and a correlated noise term, i.e., $\theta_t = 4\sin(\frac{t}{2}) + \xi_t$, and $\xi_t = \gamma\xi_{t-1} + e_t$ follows an autoregressive process with noise $e_t \sim_{iid} \mathcal{N}(0, 1)$, with known $\gamma$. The DM uses the optimal prediction $\widehat{\xi}_s^{(t)} = \gamma^{s-t+1}\xi_{t-1}$ for $s \geq t$. Further, we assume that $a_t$'s are known, and are generated such that $a_t = 1 + AB_t$ where $B_t \sim_{iid} Bern(0.3)$. We choose $A \in \{0, 50, 500\}$, where larger $A$ models ill-conditioned problems with small $\kappa$. For simplicity, we take $\mathbf{x}^{(init)} = \mathbf{0}$ or $\mathbf{x}^{(init)} = \mathbf{x}_{nf}^* := \arg\min_{\mathbf{x} \in \mathcal{X}} C(\mathbf{x}; \theta - \xi)$, the optimal solution in the *noise free* setting. For each pair of $(\gamma, A)$, we generate 100 problems, and Figure 4 plots the logarithm of the sample-average dynamics regret for $\gamma = 0.3, 0.7$, $A = 0, 500$. See Figures 5 and 6 in Appendix for more experiment results.

The results show the superior performance of online-AGM, and also the following interesting phenomenon: in addition to faster convergence of dynamic regret, online-AGM is also converging to a better point: in all settings in Figure 4, when $k = 20$, the average dynamic regret for online-AGM is strictly smaller than that of online-PGM. This might be explained by the fact that online-AGM has smaller dependency on longer-term prediction inaccuracy.

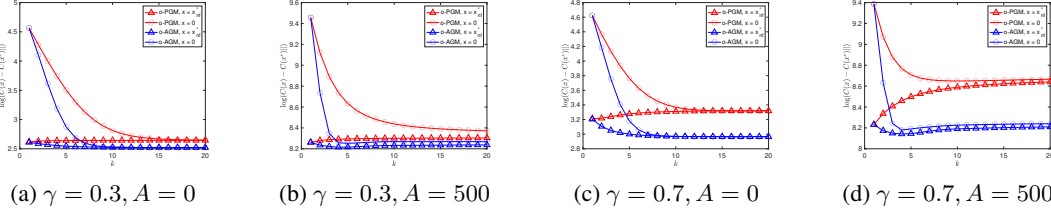

(a) $\gamma = 0.3, A = 0$      (b) $\gamma = 0.3, A = 500$      (c) $\gamma = 0.7, A = 0$      (d) $\gamma = 0.7, A = 500$

Figure 4: Logarithm of sample-average dynamic regret.

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

## A ADDITIONAL RESULTS FOR SECTION 1

### A.1 EXAMPLE 3

Consider the following problem

$$\min_{x_t \in \mathcal{X}_t,\ t \in [T]} \sum_{t=1}^{T} f_t(x_{W_t}; \xi_t), \quad s.t. \sum_{t=1}^{T} \mathbf{c}_t(x_{W_t}; \xi_t) \leq \mathbf{0}.$$

Assume that for all $t$, $\mathcal{X}_t \subset \mathbb{R}^{n_t}$, $f_t : \mathcal{X}_{W_t} \times \Xi \to \mathbb{R}$, and all components of $\mathbf{c}_t : \mathcal{X}_{W_t} \times \Xi \to \mathbb{R}^m$ are convex. This problem has the Lagrangian

$$\mathcal{L}(\mathbf{x}, \boldsymbol{\lambda}; \boldsymbol{\xi}) := \sum_{t=1}^{T} f_t(x_{W_t}; \xi_t) + \boldsymbol{\lambda}^T \sum_{t=1}^{T} \mathbf{c}_t(x_{W_t}; \xi_t).$$

Under strong duality, the original problem is equivalent to minimizing $\mathcal{L}(\mathbf{x}, \boldsymbol{\lambda}^*(\boldsymbol{\xi}); \boldsymbol{\xi})$ where $\boldsymbol{\lambda}^*(\boldsymbol{\xi})$ is the optimal dual (which might depend on $\boldsymbol{\xi}$). Thus, we can define $\theta_t = (\xi_t, \lambda_t)$ with $\theta_t^* = (\xi_t^*, \boldsymbol{\lambda}^*(\boldsymbol{\xi}^*))$. Then we have

$$\mathcal{L}(\mathbf{x}, \boldsymbol{\lambda}^*(\boldsymbol{\xi}^*); \boldsymbol{\xi}^*) = \sum_{t=1}^{T} \tilde{f}_t(x_t; \theta_t^*), \quad \tilde{f}_t(x_{W_t}; \theta_t) := f_t(x_{W_t}; \xi_t) + \lambda_t^T \mathbf{c}_t(x_{W_t}; \xi_t).$$

Predictions of $\xi_t$ and $\boldsymbol{\lambda}^*(\boldsymbol{\xi})$ might be available from prior information or past data about the model.

## A.2 Comment on Assumption 1.2

In Assumption 1.2, we assume that $C(\cdot; \boldsymbol{\theta}) : \mathcal{X} \to \mathbb{R}$ is 1-smooth and $\kappa$-strongly convex. We can make the following connection with the assumption that for each $t$, there exists $0 \le \alpha_t \le \beta_t$, such that $f_t : \mathcal{X}_{W_t} \to \mathbb{R}$ is $\alpha_t$-strongly convex and $\beta_t$-smooth, i.e. for all $x, y \in \mathcal{X}_{W_t}$,

$$\frac{\alpha_t}{2} \|y - x\|^2 \le f_t(y; \theta_t) - f_t(x; \theta_t) - \langle \nabla f_t(x; \theta_t), y - x \rangle \le \frac{\beta_t}{2} \|y - x\|^2.$$

For all $\mathbf{x}, \mathbf{y} \in \mathcal{X}$, adding the above inequalities for all $t$, we get

$$\sum_{t=1}^{T} \frac{\alpha_t}{2} \|y_{W_t} - x_{W_t}\|^2 \le C(\mathbf{y}; \boldsymbol{\theta}) - C(\mathbf{x}; \boldsymbol{\theta}) - \langle \nabla C(\mathbf{x}; \boldsymbol{\theta}), \mathbf{y} - \mathbf{x} \rangle \le \sum_{t=1}^{T} \frac{\beta_t}{2} \|y_{W_t} - x_{W_t}\|^2.$$

Since $W_t = [\max(1, t - a), \min(t + b, T)]$,

$$\sum_{t=1}^{T} \frac{\beta_t}{2} \|x_{W_t} - y_{W_t}\|^2 = \sum_{t=1}^{T} \sum_{s \in W_t} \frac{\beta_t}{2} \|x_s - y_s\|^2 = \sum_{s=1}^{T} \sum_{t=\max(1, s-b)}^{\min(s+a, T)} \frac{\beta_t}{2} \|x_s - y_s\|^2 \le \frac{B}{2} \|\mathbf{x} - \mathbf{y}\|^2,$$

where $B = \max_{s \in [T]} \sum_{t=\max(1,s-b)}^{\min(s+a,T)} \beta_t$. Similarly, we can show that

$$\sum_{t=1}^{T} \frac{\alpha_t}{2} \|x_{W_t} - y_{W_t}\|^2 \ge \frac{A}{2} \|\mathbf{x} - \mathbf{y}\|^2, \quad A = \min_{s \in [T]} \sum_{t=\max(1,s-b)}^{\min(s+a,T)} \alpha_t.$$

When $\beta_t \le \overline{\beta}$ and $\alpha_t \ge \overline{\alpha}$ for all $t$, we have $A \ge (1 + \min(a, b))\overline{\alpha}$, and $B \le (1 + a + b)\overline{\beta}$. That is, the normalized objective, $\frac{C}{(1+a+b)\overline{\beta}}$, is 1-smooth and $\frac{(1+\min(a,b))\overline{\alpha}}{(1+a+b)\overline{\beta}}$-strongly convex. In particular, $\kappa = \frac{(1+\min(a,b))\overline{\alpha}}{(1+a+b)\overline{\beta}}$ does not depend on $T$ when $a$ and $b$ are on the same scale. In fact, from the expressions for $A$ and $B$, we see that it's possible that $\alpha_t = 0$ for some $t$, but $A > 0$, i.e. $f_t$ is not strongly convex, but $C$ still is. Thus, our Assumption 1.2 can be seen as weaker than assumptions on each $f_t$.

# B Additional Results for Section 3.1

**Proposition B.1** (Devolder et al. (2013b)Devolder et al. (2013a)). *Let the oracle* $\tilde{\mathcal{O}}(y) = (\tilde{F}(y), \tilde{\nabla} F(y))$ *such that for all* $y \in \mathcal{K}$, $|F(y) - \tilde{F}(y)| \le \Delta_1$, *and* $\|\nabla F(y) - \tilde{\nabla} F(y)\| \le \Delta_2$.
- *If* $F$ *has a* $(0, 0, M')$-*oracle, i.e.* $F$ *is* $M'$-*smooth, and* $\mathcal{K}$ *is bounded and has diameter* $D_{\mathcal{K}}$, *then* $(\tilde{F}(y) - \Delta_1 - \Delta_2 D_{\mathcal{K}}, \tilde{\nabla} F(y))$ *is a* $(2\Delta_1 + 2\Delta_2 D_{\mathcal{K}}, 0, M')$-*oracle.*
- *If* $F$ *has a* $(0, m', M')$-*oracle for some* $m' > 0$, *i.e.* $F$ *is* $M'$-*smooth and* $m'$-*strongly convex, then* $(\tilde{F}(y) - \Delta_1 - \frac{\Delta_2^2}{m'}, \tilde{\nabla} F(y))$ *is a* $(2\Delta_1 + (\frac{1}{m'} + \frac{1}{2M})\Delta_2^2, \frac{m'}{2}, 2M')$-*oracle.*

**Projected gradient method.** Projected gradient method with inexact first order gradient oracle is the same as the classical PGM, but using the inexact gradient returned by an $(\delta_l, m, M)$-oracle $O^{(l)}$. That is, at the $l$-th iteration,

$$(f^{(l-1)}, g^{(l-1)}) \leftarrow \mathcal{O}^{(l)}(x^{(l-1)}), \quad x^{(l)} \leftarrow \mathsf{Proj}_{\mathcal{K}}(x^{(l-1)} - \frac{1}{M} g^{(l-1)}). \tag{14}$$

The exact algorithm is presented in Algorithm 1. Below we state its convergence properties:

**Theorem B.1** (Theorem 2 in Devolder et al. (2013b), modification of Theorem 4 in Devolder et al. (2013a))**.** *The sequence generated by Algorithm 1, with a sequence of $(\delta_l, m, M)$-oracle $\mathcal{O}^{(l)}$ for $l \in \mathbb{N}$, satisfies the following properties:*

$$
F(\overline{x}^{(l)}) - F(x^*) \leq
\begin{cases}
\frac{M}{2} \exp(-\frac{m}{M}l)\|x^{(0)} - x^*\|^2 + \quad \frac{m/M}{1-(1-\frac{m}{M})^l} \sum_{i=1}^{l}(1 - \frac{m}{M})^{l-i}\delta_i \\
\qquad\qquad\qquad\qquad\qquad (m > 0,\; \overline{x}^{(l)} := \frac{m/M}{1-(1-\frac{m}{M})^l} \sum_{i=1}^{l}(1 - \frac{m}{M})^{l-i}x^{(i)}) \\
\frac{M}{2l}\|x^{(0)} - x^*\|^2 + \frac{1}{l}\sum_{i=1}^{l}\delta_i \quad (m = 0,\; \overline{x}^{(l)} := \frac{1}{l}\sum_{i=1}^{l}x^{(i)})
\end{cases}
\tag{15}
$$

**Accelerated Gradient Method.** To obtain the accelerated convergence rate, Algorithm 2 performs projected gradient descent at a extrapolated point:

$$
(f^{(l)}, g^{(l)}) \leftarrow \mathcal{O}^{(l)}(y^{(l)}), \quad x^{(l)} \leftarrow \mathsf{Proj}_{\mathcal{K}}(y^{(l)} - \frac{1}{M}g^{(l)}).
\tag{16}
$$

After that, the following updates are used, where $(\tau_l)_{l \in \mathbb{N}}$ is a sequence that depends on $m, M$:

$$
v^{(l)} \leftarrow v^{(l-1)} + \alpha_l(g^{(l)} - my^{(l)}), \quad z^{(l)} \leftarrow \mathsf{Proj}_{\mathcal{K}}(\frac{My^{(1)} - v^{(l)}}{m(\sum_{j=1}^{l}\alpha_j) + M}), \quad y^{(l+1)} \leftarrow \tau_l z^{(l)} + (1 - \tau_l)x^{(l)}
\tag{17}
$$

**Theorem B.2** (modification of Theorem 6 in Devolder et al. (2013a))**.** *For Algorithm 2, choose the sequences $(\alpha_l)_{l \in \mathbb{N}}$ and $(\tau_l)_{l \in \mathbb{N}}$ such that $\alpha_1 = 1$, $1 + \frac{m}{M}A_l = \frac{\alpha_{l+1}^2}{A_{l+1}}$ for all $l \in \mathbb{N}$, where $A_l := \sum_{i=1}^{l}\alpha_i$, and $\tau_l = \frac{\alpha_{l+1}}{A_{l+1}}$, then for all $l \in \mathbb{N}$, we have*

$$
F(x^{(l)}) - F(x^*) \leq
\begin{cases}
(1 + \frac{1}{2}\sqrt{\frac{m}{M}})^{-2l} \cdot 3M\|x^{(init)} - x^*\|^2 + \sum_{i=1}^{l}(1 + \frac{1}{2}\sqrt{\frac{m}{M}})^{-2(l-i)}\delta_i & m > 0 \\
\frac{4M}{l^2}\|x^{(init)} - x^*\|^2 + 4\sum_{i=1}^{l}(\frac{i}{l})^2\delta_i & m = 0
\end{cases}
\tag{18}
$$

In Devolder et al. (2013b), the authors study inexact first oracle when the objective function is smooth but not strongly convex. In Devolder et al. (2013a) the authors study the strongly convex case with inexact oracle, but $\delta_l$'s are the same for all time period. The proof of Theorem B.1 and B.2 are adaptation of the proof in Devolder et al. (2013a) for the strongly convex and smooth objective functions, taking into account variation in $\delta_l$ across time.

## B.1 Projected Gradient Method

*Proof of Theorem B.1.* The case when $m = 0$ follows directly from Theorem 2 in Devolder et al. (2013b). Below, we adapt the proof of Theorem 6 in Devolder et al. (2013a) to show the bound when $m > 0$.

For convenience, denote $r_l^2 = \|x^{(l)} - x^*\|^2$ for $l = 0, 1, 2, \ldots$. Then we have

$$
r_{l+1}^2 = \|x^{(l+1)} - x^*\|^2 = r_l^2 + 2\langle x^{(l+1)} - x^{(l)}, x^{(l+1)} - x^* \rangle - \|x^{(l+1)} - x^{(l)}\|^2.
$$

The optimality condition at $x^{(l+1)}$ implies that

$$
\langle g^{(l)} + M(x^{(l+1)} - x^{(l)}), x - x^{(l+1)} \rangle \geq 0, \quad \forall x \in \mathcal{K}.
$$

Thus we have

$$
\langle x^{(l+1)} - x^{(l)}, x^{(l+1)} - x^* \rangle \leq \frac{1}{M}\langle g^{(l)}, x^* - x^{(l+1)} \rangle.
$$

Thus we have

$$
\begin{aligned}
r_{l+1}^2 &\leq r_l^2 + \frac{2}{M}\langle g^{(l)}, x^* - x^{(l+1)} \rangle - \|x^{(l+1)} - x^{(l)}\|^2 \\
&= r_l^2 + \frac{2}{M}\langle g^{(l)}, x^* - x^{(l)} \rangle - \frac{2}{M}[\langle g^{(l)}, x^{(l+1)} - x^{(l)} \rangle + \frac{M}{2}\|x^{(l+1)} - x^{(l)}\|^2] \\
&\leq r_l^2 + \frac{2}{M}\langle g^{(l)}, x^* - x^{(l)} \rangle - \frac{2}{M}[F(x^{(l+1)}) - f^{(l)} - \delta_{l+1}] \quad (\mathcal{O}^{(l+1)} \text{ is } (\delta_{l+1}, m, M) \text{ oracle}) \\
&\leq r_l^2 + \frac{2}{M}[F(x^*) - f^{(l)} - \frac{m}{2}\|x^* - x^{(l)}\|^2] - \frac{2}{M}[F(x^{(l+1)}) - f^{(l)} - \delta_{l+1}] \\
&= (1 - \frac{m}{M})r_l^2 + \frac{2}{M}[F(x^*) - F(x^{(l+1)}) + \delta_{l+1}].
\end{aligned}
$$

Denoting $R_l := \frac{2}{M}[F(x^*) - F(x^{(l+1)}) + \delta_{l+1}]$, and applying the above bound recursively, we get

$$r_l^2 \leq (1 - \frac{m}{M})r_{l-1}^2 + R_{l-1} \leq (1 - \frac{m}{M})^l r_0^2 + \sum_{i=1}^{l}(1 - \frac{m}{M})^{i-1} R_{l-i}.$$

Thus we have

$$0 \leq (1 - \frac{m}{M})^l r_0^2 + \sum_{i=1}^{l}(1 - \frac{m}{M})^{i-1} R_{l-i},$$

which implies that

$$\sum_{i=1}^{l}(1 - \frac{m}{M})^{l-i}[F(x^{(i)}) - F(x^*)] \leq (1 - \frac{m}{M})^l \cdot \frac{M}{2} r_0^2 + \sum_{i=1}^{l}(1 - \frac{m}{M})^{l-i}\delta_i.$$

$\square$

## B.2 ACCELERATED GRADIENT METHOD

**Lemma B.1.** *Denote*

$$\psi_l^* := \min_{x \in \mathcal{K}} \frac{M}{2}\|x - y^{(1)}\|^2 + \sum_{i=1}^{l} \alpha_i[f^{(i)} + \langle g^{(i)}, x - y^{(i)}\rangle + \frac{m}{2}\|x - y^{(i)}\|^2].$$

*Choosing the sequence $(\alpha_l)_{l \in \mathbb{N}}$ and sequence $(\tau_l)_{l \in \mathbb{N}}$ such that*

$$1 + \frac{m}{M}A_l = \frac{\alpha_{l+1}^2}{A_{l+1}}, \quad \alpha_1 = 1$$

*where $A_l := \sum_{i=1}^{l} \alpha_i$, and $\tau_l = \frac{\alpha_{l+1}}{A_{l+1}}$. Then for any $l \geq 1$, we have $A_l F(x^{(l)}) \leq \psi_l^* + E_l$ where $E_l = \sum_{i=1}^{l} A_i \delta_i$.*

*Proof of Lemma B.1.* The proof is an adaptation of the proof for Lemma 3 in Devolder et al. (2013a). The proof is by induction on $l$. For $l = 1$,

$$\psi_1^* = \min_{x \in \mathcal{K}} \frac{M}{2}\|y - y^{(1)}\|^2 + f^{(1)} + \langle g^{(1)}, x - y^{(1)}\rangle + \frac{m}{2}\|x - y^{(1)}\|^2 \quad \text{(using } \alpha_1 = 1\text{)}$$

$$\geq \min_{x \in \mathcal{K}} f^{(1)} + \langle g^{(1)}, x - y^{(1)}\rangle + \frac{M}{2}\|x - y^{(1)}\|^2$$

$$= f^{(1)} + \langle g^{(1)}, x^{(1)} - y^{(1)}\rangle + \frac{M}{2}\|x^{(1)} - y^{(1)}\|^2 \quad \text{(by the update rule)}$$

$$\geq F(x^{(1)}) - \delta_1. \quad (\mathcal{O}^{(1)} \text{ is } (\delta_1, m, M)\text{-oracle})$$

Thus the statement is true for $l = 1$. Suppose that it's true for some $l \geq 1$, then the optimality condition for $z^{(l)}$ implies that

$$\langle M(z^{(l)} - y^{(1)}) + v^{(l)} + mA_l z^{(l)}, y - z^{(l)}\rangle \geq 0, \quad \forall y \in \mathcal{K}.$$

By the update rule,

$$v^{(l)} = \sum_{i=1}^{l} \alpha_i(g^{(i)} - my^{(i)}) \implies v^{(l)} + mA_l z^{(l)} = \sum_{i=1}^{l} \alpha_i(g^{(i)} - my^{(i)} + mz^{(l)}),$$

and so we get

$$M\langle z^{(l)} - y^{(1)}, y - z^{(l)}\rangle \geq \langle \sum_{i=1}^{l} \alpha_i(g^{(i)} - my^{(i)} + mz^{(l)}), z^{(l)} - y\rangle, \quad \forall y \in \mathcal{K}.$$

Since $\|x - y^{(1)}\|^2$ is strongly convex in $x$, we have for any $x \in \mathcal{K}$,

$$\frac{M}{2}\|x - y^{(1)}\|^2 - \frac{M}{2}\|z^{(l)} - y^{(1)}\|^2 \geq M\langle z^{(l)} - y^{(1)}, x - z^{(l)}\rangle + \frac{M}{2}\|x - z^{(l)}\|^2$$

$$\geq \langle \sum_{i=1}^{l} \alpha_i(g^{(i)} - my^{(i)} + mz^{(l)}), z^{(l)} - x\rangle + \frac{M}{2}\|x - z^{(l)}\|^2$$

$$= \sum_{i=1}^{l} \alpha_i\langle g^{(i)}, z^{(l)} - x\rangle + m\sum_{i=1}^{l} \alpha_i\langle -y^{(i)} + z^{(l)}, z^{(l)} - x\rangle + \frac{M}{2}\|x - z^{(l)}\|^2.$$

Thus for any $x \in \mathcal{K}$, for the objective function in the definition of $\psi_{l+1}^*$,

$$\frac{M}{2}\|x - y^{(1)}\|^2 + \sum_{i=1}^{l+1} \alpha_i[f^{(i)} + \langle g^{(i)}, x - y^{(i)}\rangle + \frac{m}{2}\|x - y^{(i)}\|^2]$$

$$\geq \frac{M}{2}\|z^{(l)} - y^{(1)}\|^2 + \sum_{i=1}^{l} \alpha_i[f^{(i)} + \langle g^{(i)}, x - y^{(i)}\rangle + \frac{m}{2}\|x - y^{(i)}\|^2]$$

$$+ \sum_{i=1}^{l} \alpha_i\langle g^{(i)}, z^{(l)} - x\rangle + m\sum_{i=1}^{l} \alpha_i\langle -y^{(i)} + z^{(l)}, z^{(l)} - x\rangle + \frac{M}{2}\|x - z^{(l)}\|^2$$

$$+ \alpha_{l+1}[f^{(l+1)} + \langle g^{(l+1)}, x - y^{(l+1)}\rangle + \frac{m}{2}\|x - y^{(l+1)}\|^2]$$

$$= \frac{M}{2}\|z^{(l)} - y^{(1)}\|^2 + \sum_{i=1}^{l} \alpha_i[f^{(i)} + \langle g^{(i)}, z^{(l)} - y^{(i)}\rangle + \frac{m}{2}\|x - y^{(i)}\|^2]$$

$$+ m\sum_{i=1}^{l} \alpha_i\langle -y^{(i)} + z^{(l)}, z^{(l)} - x\rangle + \frac{M}{2}\|x - z^{(l)}\|^2$$

$$+ \alpha_{l+1}[f^{(l+1)} + \langle g^{(l+1)}, x - y^{(l+1)}\rangle + \frac{m}{2}\|x - y^{(l+1)}\|^2].$$

Using the following relation,

$$\langle z^{(l)} - y^{(i)}, z^{(l)} - x\rangle = \frac{1}{2}\|z^{(l)} - y^{(i)}\|^2 + \frac{1}{2}\|z^{(l)} - x\|^2 - \frac{1}{2}\|x - y^{(i)}\|^2,$$

we get

$$\frac{M}{2}\|x - y^{(1)}\|^2 + \sum_{i=1}^{l+1} \alpha_i[f^{(i)} + \langle g^{(i)}, x - y^{(i)}\rangle + \frac{m}{2}\|x - y^{(i)}\|^2]$$

$$\geq \frac{M}{2}\|z^{(l)} - y^{(1)}\|^2 + \sum_{i=1}^{l} \alpha_i[f^{(i)} + \langle g^{(i)}, z^{(l)} - y^{(i)}\rangle + \frac{m}{2}\|x - y^{(i)}\|^2]$$

$$+ \frac{m}{2}\sum_{i=1}^{l} \alpha_i(\|z^{(l)} - y^{(i)}\|^2 + \|z^{(l)} - x\|^2 - \|x - y^{(i)}\|^2) + \frac{M}{2}\|x - z^{(l)}\|^2$$

$$+ \alpha_{l+1}[f^{(l+1)} + \langle g^{(l+1)}, x - y^{(l+1)}\rangle + \frac{m}{2}\|x - y^{(l+1)}\|^2]$$

$$= \frac{M}{2}\|z^{(l)} - y^{(1)}\|^2 + \sum_{i=1}^{l} \alpha_i[f^{(i)} + \langle g^{(i)}, z^{(l)} - y^{(i)}\rangle + \frac{m}{2}\|z^{(l)} - y^{(i)}\|^2] + \frac{M + mA_l}{2}\|x - z^{(l)}\|^2$$

$$+ \alpha_{l+1}[f^{(l+1)} + \langle g^{(l+1)}, x - y^{(l+1)}\rangle + \frac{m}{2}\|x - y^{(l+1)}\|^2]$$

$$= \psi_l^* + \frac{M + mA_l}{2}\|x - z^{(l)}\|^2 + \alpha_{l+1}[f^{(l+1)} + \langle g^{(l+1)}, x - y^{(l+1)}\rangle + \frac{m}{2}\|x - y^{(l+1)}\|^2]$$

Thus we have

$$\psi_{l+1}^* \geq \psi_l^* + \min_{x \in \mathcal{K}} \frac{M + mA_l}{2} \|x - z^{(l)}\|^2 + \alpha_{l+1}[f^{(l+1)} + \langle g^{(l+1)}, x - y^{(l+1)} \rangle + \frac{m}{2} \|x - y^{(l+1)}\|^2].$$

By induction $A_l F(x^{(l)}) \leq \psi_l^* + E_l$, and so

$$\psi_l^* + \alpha_{l+1}[f^{(l+1)} + \langle g^{(l+1)}, x - y^{(l+1)} \rangle + \frac{m}{2} \|x - y^{(l+1)}\|^2]$$

$$\geq A_l F(x^{(l)}) - E_l + \alpha_{l+1}[f^{(l+1)} + \langle g^{(l+1)}, x - y^{(l+1)} \rangle + \frac{m}{2} \|x - y^{(l+1)}\|^2]$$

$$\geq A_l[f^{(l+1)} + \langle g^{(l+1)}, x^{(l)} - y^{(l+1)} \rangle + \frac{m}{2} \|x^{(l)} - y^{(l+1)}\|^2] - E_l$$

$$+ \alpha_{l+1}[f^{(l+1)} + \langle g^{(l+1)}, x - y^{(l+1)} \rangle + \frac{m}{2} \|x - y^{(l+1)}\|^2]$$

$$= A_{l+1} f^{(l+1)} + \langle g^{(l+1)}, A_l(x^{(l)} - y^{(l+1)}) + \alpha_{l+1}(y - y^{(l+1)}) \rangle - E_l$$

$$+ \frac{A_l m}{2} \|x^{(l)} - y^{(l+1)}\|^2 + \frac{\alpha_{l+1} m}{2} \|x - y^{(l+1)}\|^2.$$

Since $\tau_l = \frac{\alpha_{l+1}}{A_{l+1}}$, and $y^{(l+1)} = \tau_l z^{(l)} + (1 - \tau_l) x^{(l)}$,

$$A_l(x^{(l)} - y^{(l+1)}) + \alpha_{l+1}(y - y^{(l+1)}) = \alpha_{l+1}(y - z^{(l)}).$$

Thus we get

$$\psi_l^* + \alpha_{l+1}[f^{(l+1)} + \langle g^{(l+1)}, x - y^{(l+1)} \rangle + \frac{m}{2} \|x - y^{(l+1)}\|^2]$$

$$\geq A_{l+1} f^{(l+1)} - E_l + \alpha_{l+1} \langle g^{(l+1)}, y - z^{(l)} \rangle.$$

Thus for the $\psi_{l+1}^*$, since we choose the sequence such that $\tau_l = \frac{\alpha_{l+1}}{A_{l+1}}$ and $A_{l+1} \tau_l^2 = 1 + \frac{m}{M} A_l$

$$\psi_{l+1}^* \geq A_{l+1} f^{(l+1)} - E_l + \min_{x \in \mathcal{K}} \frac{M + mA_l}{2} \|x - z^{(l)}\|^2 + \alpha_{l+1} \langle g^{(l+1)}, y - z^{(l)} \rangle$$

$$= -E_l + A_{l+1}[f^{(l+1)} + \min_{x \in \mathcal{K}} \frac{\tau_l^2 M}{2} \|x - z^{(l)}\|^2 + \tau_l \langle g^{(l+1)}, x - z^{(l)} \rangle]$$

For $x \in \mathcal{K}$, define $\widehat{x} = \tau_l x + (1 - \tau_l) x^{(l)}$, since $\tau_l(x - z^{(l)}) = \widehat{x} - y^{(l+1)}$,

$$\min_{x \in \mathcal{K}} \frac{\tau_l^2 M}{2} \|x - z^{(l)}\|^2 + \tau_l \langle g^{(l+1)}, x - z^{(l)} \rangle$$

$$= \min_{\widehat{x} \in \tau_l \mathcal{K} + (1 - \tau_l) x^{(l)}} \frac{M}{2} \|\widehat{x} - y^{(l+1)}\|^2 + \langle g^{(l+1)}, \widehat{x} - y^{(l+1)} \rangle$$

$$\geq \min_{\widehat{x} \in \mathcal{K}} \frac{M}{2} \|\widehat{x} - y^{(l+1)}\|^2 + \langle g^{(l+1)}, \widehat{x} - y^{(l+1)} \rangle$$

Putting the above two equations together, we get

$$\psi_{l+1}^* \geq -E_l + A_{l+1}[f^{(l+1)} + \min_{\widehat{x} \in \mathcal{K}} \frac{M}{2} \|\widehat{x} - y^{(l+1)}\|^2 + \langle g^{(l+1)}, \widehat{x} - y^{(l+1)} \rangle]$$

$$\geq A_{l+1} F(x^{(l+1)}) - E_l - A_{l+1} \delta_{l+1}.$$

where the last step uses $\mathcal{O}^{(l+1)}$ is $(\delta_{l+1}, m, M)$-oracle.

$\square$

*Proof of Theorem B.2.*

$$\psi_l^* = \min_{x \in \mathcal{K}} \frac{M}{2} \|x - y^{(1)}\|^2 + \sum_{i=1}^{l} \alpha_i[f^{(i)} + \langle g^{(i)}, x - y^{(i)} \rangle + \frac{m}{2} \|x - y^{(i)}\|^2]$$

$$\leq \frac{M}{2} \|x^* - y^{(1)}\|^2 + \sum_{i=1}^{l} \alpha_i[f^{(i)} + \langle g^{(i)}, x^* - y^{(i)} \rangle + \frac{m}{2} \|x^* - y^{(i)}\|^2]$$

$$\leq \frac{M}{2} \|x^* - y^{(1)}\|^2 + A_l F(x^*),$$

where the last step is because $\mathcal{O}^{(i)}$ is $(\delta_i, m, M)$-oracle. Since $y^{(1)} = x^{(init)}$, together with Lemma B.1, we have

$$F(x^{(l)}) \leq \frac{M}{2A_l}\|x^{(init)} - x^*\|^2 + F(x^*) + \sum_{i=1}^{l} \frac{A_i}{A_l}\delta_i.$$

When $m > 0$, from the proof of Lemma 4 in Devolder et al. (2013a), $A_{k+1} \geq (1 + \frac{1}{2}\sqrt{m/M})^2 A_k$ for all $k \geq 1$, giving the desired bound.

When $m = 0$, Remark 10 in Devolder et al. (2013a) shows that $A_k \geq (k+1)^2/4$ for all $k \geq 1$. In addition, $\alpha_1 = A_1 = 1$, we can use induction to show that $A_k \leq k^2$ for all $k$ and $\alpha_k \leq k$ for all $k$: $\alpha_{k+1}^2 = \alpha_{k+1} + A_k$ and so $\alpha_{k+1} = 1/2 + \sqrt{1/4 + A_k} \leq k+1$, $A_{k+1} \leq k^2 + k + 1 \leq (k+1)^2$. Thus, $A_i/A_l \leq 4i^2/(l+1)^2$. $\qquad\square$

# C ADDITIONAL RESULTS FOR SECTION 3.2

## C.1 ADDITIONAL RESULTS FOR THE HYPOTHETICAL OFFLINE PROBLEM

**Proposition C.1.** *The oracle defined in Eq. 8 satisfies*

$$\|\nabla C(\mathbf{x};\boldsymbol{\theta}^*) - (G_1(x_{\overline{W}_1};\theta_{\widetilde{W}_1}^{(l)}), G_2(x_{\overline{W}_2};\theta_{\widetilde{W}_2}^{(l)}), \ldots, G_T(x_{\overline{W}_T};\theta_{\widetilde{W}_T}^{(l)}))\|^2 \leq \sum_{t=1}^{T}(\sum_{s\in\widetilde{W}_t} h_{s,t}\|\theta_s^* - \theta_s^{(l)}\|)^2.$$

*Proof of Proposition C.1.* By Assumption 1.1, for any $\boldsymbol{\theta} \in \Theta^T$,

$$\|\frac{\partial C}{\partial x_t}(x_{\overline{W}_t};\theta_{\widetilde{W}_t}^*) - G_t(x_{\overline{W}_t};\theta_{\widetilde{W}_t})\| = \|\sum_{s\in\widetilde{W}_t}(\frac{\partial f_s}{\partial x_t}(x_{W_s};\theta_s^*) - \frac{\partial f_s}{\partial x_t}(x_{W_s};\theta_s))\|$$

$$\leq \sum_{s\in\widetilde{W}_t}\|\frac{\partial f_s}{\partial x_t}(x_{W_s};\theta_s^*) - \frac{\partial f_s}{\partial x_t}(x_{W_s};\theta_s)\| \leq \sum_{s\in\widetilde{W}_t} h_{s,t}\|\theta_s^* - \theta_s\|.$$

With this oracle $\mathcal{O}^{(l)}$, the gradient has error

$$\|\nabla C(\mathbf{x};\boldsymbol{\theta}^*) - (G_1(x_{\overline{W}_1};\theta_{\widetilde{W}_1}^{(l)}), G_2(x_{\overline{W}_2};\theta_{\widetilde{W}_2}^{(l)}), \ldots, G_T(x_{\overline{W}_T};\theta_{\widetilde{W}_T}^{(l)}))\|^2$$

$$= \sum_{t=1}^{T}\|\frac{\partial C}{\partial x_t}(x_{\overline{W}_t};\theta_{\widetilde{W}_t}^*) - G_t(x_{\overline{W}_t};\theta_{\widetilde{W}_t}^{(l)})\|^2 \leq \sum_{t=1}^{T}(\sum_{s\in\widetilde{W}_t} h_{s,t}\|\theta_s^* - \theta_s^{(l)}\|)^2.$$

$$\square$$

For each $\mathcal{O}^{(l)}$, we can take $\Delta_1^{(l)} = 0$ and $\Delta_2^{(l)} = (\sum_{t=1}^{T}(\sum_{s\in\widetilde{W}_t} h_{s,t}\|\theta_s^* - \theta_s^{(l)}\|)^2)^{1/2}$ in Proposition B.1. Combining Proposition B.1,

**Corollary C.1.** *If $\kappa = 0$, for $l \in [L]$, $\mathcal{O}^{(l)}$ is equivalent to a $(\delta_l, 0, 1)$-oracle for $C(\cdot;\boldsymbol{\theta}^*)$, where $\delta_l = 2(\sum_{t=1}^{T}(\sum_{s\in\widetilde{W}_t} h_{s,t}\|\theta_s^* - \theta_s^{(l)}\|)^2)^{1/2}D_{\mathcal{X}}$, $D_{\mathcal{X}} = \max_{\mathbf{x},\mathbf{x}'\in\mathcal{X}}\|\mathbf{x} - \mathbf{x}'\|$ is the diameter of $\mathcal{X}$. Thus Algorithm 1 generates a sequence $x^{(1)}, x^{(2)}, \ldots, x^{(L)}$ such that*

$$C(\tilde{\mathbf{x}}^{(l)};\boldsymbol{\theta}^*) - C(\mathbf{x}^*;\boldsymbol{\theta}^*) \leq \frac{1}{2l}\|\mathbf{x}^{(init)} - \mathbf{x}^*\|^2 + \frac{1}{l}\sum_{i=1}^{l}\delta_i, \quad \tilde{\mathbf{x}}^{(l)} = \frac{1}{l}\sum_{i=1}^{l}\mathbf{x}^{(i)}.$$

*Algorithm 2 generates a sequence $x^{(1)}, x^{(2)}, \ldots, x^{(L)}$ such that*

$$C(\mathbf{x}^{(l)};\boldsymbol{\theta}^*) - C(\mathbf{x}^*;\boldsymbol{\theta}^*) \leq \frac{4}{l^2}\|\mathbf{x}^{(init)} - \mathbf{x}^*\|^2 + 4\sum_{i=1}^{l}(\frac{i}{l})^2\delta_i.$$

**Corollary C.2.** *If $\kappa > 0$, for $l \in [L]$, $\mathcal{O}^{(l)}$ is equivalent to a $(\delta_l, \frac{\kappa}{2}, 2)$-oracle for $C(\cdot; \boldsymbol{\theta}^*)$, where $\delta_l = (\frac{1}{\kappa} + \frac{1}{2})(\sum_{t=1}^{T}(\sum_{s \in \widetilde{W}_t} h_{s,t} \|\theta_s^* - \theta_s^{(l)}\|)^2)$. Thus Algorithm 1 generates a sequence $x^{(1)}, x^{(2)}, \ldots, x^{(L)}$ such that for $\overline{x}^l := \frac{\kappa/4}{1-(1-\kappa/4)^l} \sum_{i=1}^{l} (1-\kappa/4)^{l-i} \mathbf{x}^{(i)}$*

$$C(\overline{\mathbf{x}}^{(l)}; \boldsymbol{\theta}^*) - C(\mathbf{x}^*; \boldsymbol{\theta}^*) \leq \frac{\exp(-\kappa l/4)}{2} \|\mathbf{x}^{(init)} - \mathbf{x}^*\|^2 + \frac{\kappa/4}{1-(1-\kappa/4)^l} \sum_{i=1}^{l} (1-\kappa/4)^{l-i} \delta_i$$

*Algorithm 2 generates a sequence $x^{(1)}, x^{(2)}, \ldots, x^{(L)}$ such that*

$$C(\mathbf{x}^{(l)}; \boldsymbol{\theta}^*) - C(\mathbf{x}^*; \boldsymbol{\theta}^*) \leq 6(1 + \frac{1}{4}\sqrt{\kappa})^{-2l} \|\mathbf{x}^{(init)} - \mathbf{x}^*\|^2 + \sum_{i=1}^{l} (1 + \frac{1}{4}\sqrt{\kappa})^{-2(l-i)} \delta_i.$$

## C.2 ADDITION RESULTS FOR THE UPDATE RULE

---

**Algorithm 4:** synchronous update

---

**Input:** $G = (V, E)$ the underlying graph, $\phi_v : \mathcal{S}_{\overline{\mathcal{N}}(v)} \to \mathcal{S}_v$ the state transition function and $s_v^{(0)} \in \mathcal{S}_v$ the initial state for all $v \in V$, $L$ the number of updates.

**Output:** $s_v^{(L)}$ for all $v \in V$.

**for** $l = 1, 2, \ldots, L$ **do**

   **for** $v \in V$ **do**           `// update order for `$v$` does not matter`

      update $s_v^{(l)} \leftarrow \phi_v(s_{\overline{\mathcal{N}}(v)}^{(l-1)})$

   **end**

**end**

---

---

**Algorithm 5:** asynchronous update

---

**Input:** $G = (V, E)$ the underlying graph, $\phi_v : \mathcal{S}_{\overline{\mathcal{N}}(v)} \to \mathcal{S}_v$ the state transition function and $s_v^{(0)} \in \mathcal{S}_v$ the initial state for all $v \in V$, $L$ the number of updates, $\sigma : [|V|] \to V$ the output order.

**Output:** $\tilde{s}_{\sigma(1)}^{(L)}, \tilde{s}_{\sigma(2)}^{(L)}, \ldots, \tilde{s}_{\sigma(|V|)}^{(L)}$

initialize $\mathcal{H} = \emptyset$;    `// `$\mathcal{H} \subset (\{0\} \cup [L]) \times V$` contains pairs `$(l,v)$` s.t. `$\tilde{s}_v^{(l)}$` has been computed`

**for** $i = 1, 2, \ldots, |V|$ **do**                                   `// compute `$\tilde{s}_{\sigma(i)}^{(L)}$

   **for** $v \in \overline{\mathcal{N}}_L(\sigma(i)) \setminus \cup_{j=1}^{i-1} \overline{\mathcal{N}}_L(\sigma(j))$ **do**

      initialize $\tilde{s}_v^{(0)} \leftarrow s_v^{(0)}$, $\mathcal{H} \leftarrow \mathcal{H} \cup \{(0, v)\}$;

   **end**

   **for** $l = 1, 2, \ldots, L$ **do**          `// compute `$\tilde{s}_{\overline{\mathcal{N}}_{L-l}(\sigma(i))}^{(l)}$` using `$\tilde{s}_{\overline{\mathcal{N}}_{L-l+1}(\sigma(i))}^{(l-1)}$

      **for** $v \in \overline{\mathcal{N}}_{L-l}(\sigma(i))$ **do**     `// update order for `$v$` does not matter`

         **if** $(l, v) \notin \mathcal{H}$ **then**         `// `$\tilde{s}_v^{(l)}$` has not been computed yet`

            update $\tilde{s}_v^{(l)} \leftarrow \phi_v(\tilde{s}_{\overline{\mathcal{N}}(v)}^{(l-1)})$, $\mathcal{H} \leftarrow \mathcal{H} \cup \{(l, v)\}$;

        **end**

      **end**

   **end**

**end**

---

**Theorem C.1.** *The update rules in Algorithm 5 are valid. Thus computation of $\tilde{s}_{\sigma(i)}^{(1:L)}$ does not require knowledge about $s_u^{(0)}$ for any $u \notin \cup_{j=1}^{i} s_{\overline{\mathcal{N}}_L(\sigma(j))}^{(0)}$.*

*In addition, given the same set of inputs (graph $G$, state transition functions $\phi_v$, and initial states $s_v^{(0)}$), for any order of output $\sigma : [|V|] \to V$, Algorithm 5 and Algorithm 4 produce the same state evolution: $s_v^{(l)} = \tilde{s}_v^{(l)}$ for all $v \in V$, $l \in [L]$.*

*Proof of Theorem C.1.* For convenience, we denote the state of the set $\mathcal{H}$ in the $i$-th outer iteration, right *after* the initialization of $\tilde{s}_v^{(0)}$, as $\mathcal{H}_{i,0}$; right *after* the $l$-th inner iteration, as $\mathcal{H}_{i,l}$ for $l = 1, 2, \ldots, L$.

We claim that

- valid update rule: for each $i \in [|V|]$ and $l \in [L]$, before entering the $l$-th iteration, $\tilde{s}_{\overline{\mathcal{N}}_{L-l+1}(\sigma(i))}^{(l-1)}$ has been computed, i.e. $(l-1, v) \in \mathcal{H}_{i,l-1}$, for all $v \in \overline{\mathcal{N}}_{L-l+1}(\sigma(i))$;

- consistent output: for all $l \in \{0, 1, \ldots, L\}$, $i \in [|V|]$, for all $(l', v') \in \mathcal{H}_{i,l}$, $s_{v'}^{(l')} = \tilde{s}_{v'}^{(l')}$.

First, it's easy to see that $\cup_{j=1}^{i} \overline{\mathcal{N}}_L(\sigma(j)) \subset \mathcal{H}_{i,0}$ for all $i \in [|V|]$.

Next we prove the first claim. For any $i \in [|V|]$, we use induction on $l$. The claim holds for $l = 1$ since $\cup_{j=1}^{i} \overline{\mathcal{N}}_L(\sigma(j)) \subset \mathcal{H}_{i,0}$, and so $(0, v) \in \mathcal{H}_{i,0}$, for all $v \in \overline{\mathcal{N}}_L(\sigma(i))$. Suppose the claim is true for some $l \leq L - 1$, then for $l + 1$, by the update process in the $l$-th iteration, for all $v \in \overline{\mathcal{N}}_{L-l}(\sigma(i))$, $\tilde{s}_v^{(l)}$ is either already computed before or is computed, and so $(l, v) \in \mathcal{H}_{i,l}$. This completes the induction.

Then we prove the second claim. The claim holds for $\mathcal{H}_{1,0}$ since $\tilde{s}_v^{(0)} = s_v^{(0)}$ for $v \in \overline{N}_L(\sigma(1))$, and $\mathcal{H}_{1,0} = \{(1, v), \ v \in \overline{N}_L(\sigma(1))\}$. Now suppose the statement holds for $\mathcal{H}_{i,l}$ for some $i \in [|V|]$ and $l \in \{0, 1, \ldots, L-1\}$, then during the $(l+1)$-th iteration, the updates are $\tilde{s}_v^{(l+1)} \leftarrow \phi_v(\tilde{s}_{\overline{\mathcal{N}}(v)}^{(l)})$, and since $\forall u \in \overline{\mathcal{N}}(v)$, $(l, u) \in \mathcal{H}_{i,l}$, by the induction hypothesis, $\tilde{s}_u^{(l)} = s_u^{(l)}$, and so $\tilde{s}_v^{(l+1)} = \phi_v(\tilde{s}_{\overline{\mathcal{N}}(v)}^{(l)}) = \phi_v(s_{\overline{\mathcal{N}}(v)}^{(l)}) = s_v^{(l+1)}$.

Suppose the claim holds for $\mathcal{H}_{i,L}$ for some $i \in [|V| - 1]$, then it holds for $\mathcal{H}_{i+1,0}$ since the only added terms are initialization $\tilde{s}_v^{(0)} = s_v^{(0)}$ for $v \in \overline{\mathcal{N}}_L(\sigma(i+1)) \setminus \cup_{j=1}^{i} \overline{\mathcal{N}}_L(\sigma(j))$.

In addition, notice that in the first $i$ iterations, $\tilde{s}_v^{(0)}$ are initialized for $v \in \cup_{j=1}^{i} \overline{\mathcal{N}}_L(\sigma(j))$ only, and the $L$ subsequent updates for $l \in [L]$ require $\tilde{s}_v^{(0)}$ but not $s_u^{(0)}$ for any $u \in V$. Thus computation of $\tilde{s}_{\sigma(i)}^{(L)}$ does not require knowledge about $s_u^{(0)}$ for any $u \notin \cup_{j=1}^{i} s_{\overline{\mathcal{N}}_L(\sigma(j))}^{(0)}$. $\square$

## C.3 STATE TRANSITION OF PGM AND AGM – A NETWORK PERSPECTIVE

Let $\mathcal{S}_t = \{0, 1, \ldots, L\} \times \mathcal{X}_t$ and $(l, x_t^{(l)}) \in \mathcal{S}_t$ represents the states of vertex $t \in V$. Since $\phi_t : \mathcal{S}_{\overline{\mathcal{N}}(t)} \to \mathcal{S}_t \cup \{\mathsf{Err}\}$ can be chosen based on Equation Eq. 9: for all $x_{\overline{W}_t} \in \mathcal{X}_{\overline{W}_t}$, $l \in [L-1]$,

$$\phi_t((l-1, x_s)_{s \in \overline{W}_t}) := (l, \mathsf{Proj}_{\mathcal{X}_t}(x_t^{(l-1)} - \frac{1}{M} G_t(x_{\overline{W}_t}^{(l-1)}; \theta_{\widetilde{W}_t}^{(l)}))),$$

and

$$\phi_t((l_s, x_s)_{s \in \overline{W}_t}) := \mathsf{Err}, \quad l_s \neq l_{s'} \text{ for some } s, s' \in \overline{W}_t, \text{ or } l_s = L \text{ for some } s \in \overline{W}_t.$$

Here the state also includes the current iteration as part of the information, and this allows us to use $l$-dependent $\theta_{\widetilde{W}_t}^{(l)}$. Also, $\phi_t$ is $\mathsf{Err}$ if the gradient $G_t$ is evaluated at different iteration-version $(l_s \neq l_{s'})$ of neighbors $x_s^{(l_s-1)}$ and $x_{s'}^{(l_{s'}-1)}$, or if $x_s$ has already been updated $L$ times. However, neither of these two cases will happen during Algorithm 5: it's easy to check that $\tilde{s}_v^{(l)} = (l, x_v^{(l)})$ is satisfied all the time, and so the input to $\phi_v$ is always of the form $(l-1, x_s)_{s \in \overline{W}_t}$ for some $l \in [L]$ and $x_{\overline{W}_t} \in \mathcal{X}_{\overline{W}_t}$.

Similarly, for Algorithm 2, the state space can be taken as $\mathcal{S}_t = \{0, 1, \ldots, L\} \times \mathcal{X}_t \times \mathbb{R}^d$ and $(l, x_t^{(l)}, \tilde{v}_t^{(l)} := v_t^{(l)} - M y_t^{(1)})$ is the state for vertex $t \in V$. In addition, since the update order only depends on the underlying graph $G = (V, E)$ which is the same for Algorithm 2 and 1, the order of

computing $s_v^{(l)}$ can be done in the same way. The difference between the two algorithms is that the (projected) gradient descent at $x_t^{(l)}$ in Algorithm 1 is replaced by (projected) gradient descent at the extrapolated point $y_t^{(l)}$, followed by an update on $\tilde{v}_t^{(l)}$ in Algorithm 2.

## C.4 CHOICE OF $L$ AND $\theta_{\widetilde{W}_v}^{(l)}$

We provide two rules for the choice of $L$ and $\theta_{\widetilde{W}_v}^{(l)}$

- requirement on $L$: notice that in iteration $i = 1, 2, \ldots, T$ of Algorithm 5, all $L$-hop neighbors of vertex $i$ must be initialized first. That is, by time $i$, $x_v^{(init)}$ must be available for all $v$ that are at most $L$-hops away from $i$, i.e. for all $v \in [T]$, such that $|v - i| \leq L(a + b)$. Thus, we choose $L(a + b) \leq k$.

- one valid choice for $\theta_{\widetilde{W}_v}^{(l)}$: if $\tilde{s}_v^l \leftarrow \phi_v(\tilde{s}_{\mathcal{N}(v)}^{(l-1)})$ is computed when computing $s_i^{(1:L)}$ (i.e. during the $i$-th outer iteration in Algorithm 5), the prediction available at time $i$ can be used, i.e. $\theta_{\widetilde{W}_v}^{(l)} = \widehat{\theta}_{\widetilde{W}_v}^{(i)}$ is a valid choice. Due to the special structure of the dependency graph for online-AGM and online-PGM – $(s, s') \in E$ if and only if $|s - s'| \leq a + b$ – the update order is also of the "fill the table" style as in Li & Li (2020) and Li et al. (2021). In particular, as presented in Table 2, we can set

$$\theta_{\widetilde{W}_v}^{(l)} = \begin{cases} \widehat{\theta}_{\widetilde{W}_v}^{(1)} & v = 1, 2, \ldots, (a+b)(L-l) + 1 \\ \widehat{\theta}_{\widetilde{W}_v}^{(v-(a+b)(L-l))} & v = (a+b)(L-l) + 2, \ldots, T \end{cases} \tag{19}$$

Table 2: choice of $\theta_{\widetilde{W}_v}^{(l)}$ in Eq. 12 for $a = 2, b = 1, L = 8$

# D ADDITIONAL SETTINGS AND RESULTS FOR NUMERICAL EXPERIMENTS

Our objective function Eq. 13 and the setting of tracking process composing of a signal term and a time-correlated noise term is a variant of the numerical experiment in Li & Li (2020), where $a_t = a$ for some $a > 0$ for all $t$. By allowing varying $a_t$, we can control the condition number of $C(\cdot; \boldsymbol{\theta})$, thereby comparing the performance of online-PGM and online-AGM for various $\kappa$'s.

We choose $T = 40$, $k \in [20]$ and $x_0 = 10$. In addition, since Algorithm 3 requires the function to be $\kappa$-strongly convex and 1-smooth, we normalize each update that involves $\nabla C$ by a factor $(2 + A)^{-1}$ and take $\kappa = \frac{2}{2+A}$[4].

For the information $\boldsymbol{\theta}^{(t)}$ at time $t$, in this experiment, $\widehat{\xi}_s^{(t)} = \xi_s$ for $s \in [t - 1]$ and $\widehat{\xi}_s^{(t)} = \gamma^{s-t+1}\xi_{t-1}$ for $s \geq t$. That is, the DM has perfect information about past $\xi_s$'s, and uses the optimal prediction (see Li & Li (2020) for more details) for prediction of unseen $\xi_s$'s.

Our experiments are run using Matlab on Macbook Pro.

In Figure 4 we provide a subset of Figures in Figure 5, and below we provide the logarithm of the average dynamic regret and the average $\|\mathbf{x} - \mathbf{x}^*\|$ for all 6 settings.

---

[4]The objective is not exactly $2 + A$-smooth and 2-strongly convex. However, this choice of parameters appear to work for this planning problem.

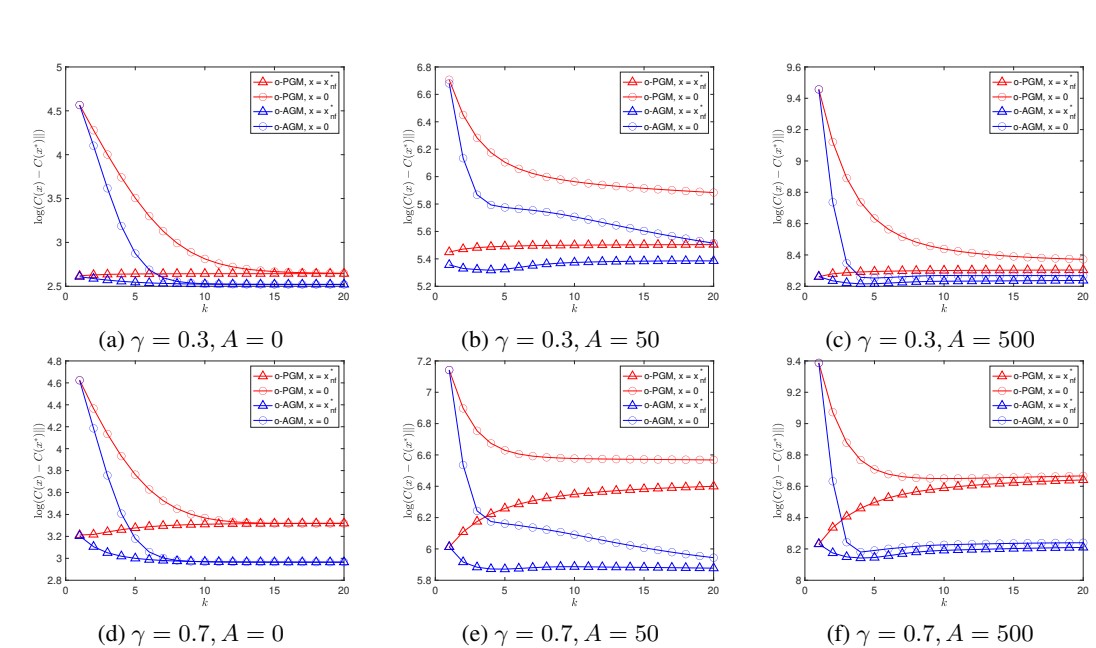

Figure 5: Logarithm of sample-average dynamic regret.

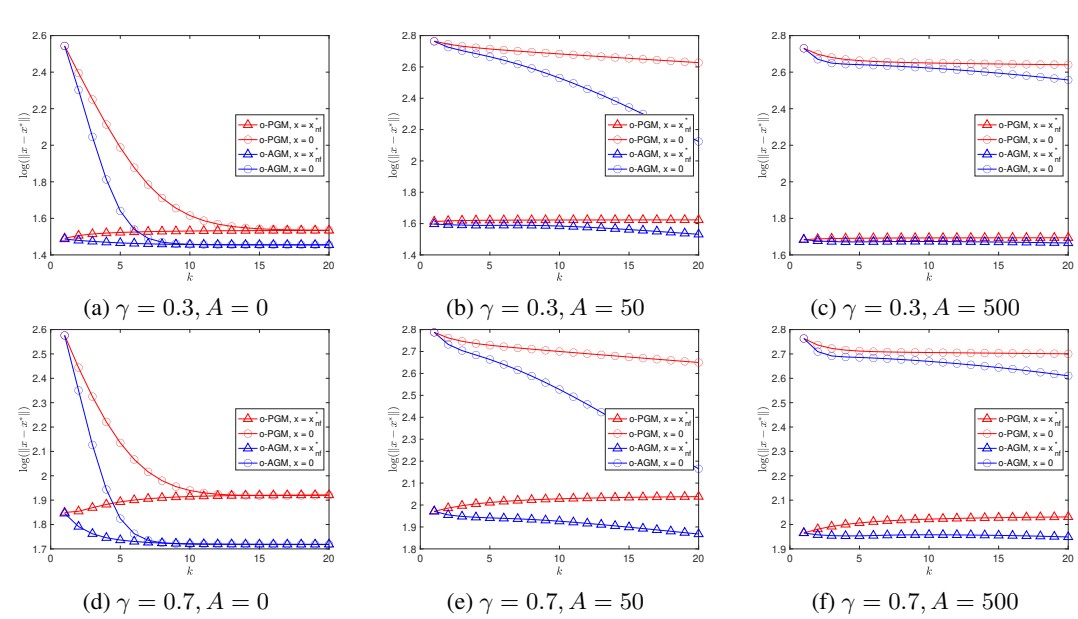

Figure 6: Logarithm of sample-average $\|\mathbf{x} - \mathbf{x}^*\|$.

