# OpenReview forum: "Online Convex Optimization with Prediction Through Accelerated Gradient Descent"
_ICLR.cc/2025/Conference — Submitted to ICLR 2025_

### Official Review · Reviewer_4fDD · 2024-10-25

**Soundness:** 3
**Presentation:** 3
**Contribution:** 2
**Rating:** 5
**Confidence:** 3

**Summary:**

This paper investigate online convex optimization where the loss $f_t$ is incurred with decision made through a time window [t-a, t+b]. This framework improves dependence on the side information accuracy and is a generalization of Li & Li (2020), thus echoes the open question among improved long term robustness on side information accuracy.

**Strengths:**

- The proposed set-up is a generalization of many problems beyond Li & Li, (2020). Which are motivated through 3 examples.
- due to the more generalized framework, the handling of unsynchronized update is more complex. Nevertheless, the paper provides detailed illustration on the update and its implication of  generalization to "fill the table" approach.
- The experimental results follows the derived bound in terms of side information prediction error and condition number

**Weaknesses:**

- One of the main claim: more robust to long term prediction error than previously established results. It would be interesting to have experimental comparison especially the experiment set-up is the same as the previous result.

- the unsynchronized update generally requires each node to wait for all necessary information from descendants and neighbours, thus the output $x_s^{l}$ is not exactly the sense of  ``online prediction'' where the decision variable $\bar{x}_{l}$ needs to be made in a timely manner regardless of the feedback. So I feel a bit concerned with the claim that this framework belongs to online prediction when motivated from the delayed decision making prospective

- it is a bit hard to established the improvement of this paper compared to previously established results, or its' optimality. Since the only trackable comparison is through motivating example 1 when $(a, b) = (1,0)$. and

**Questions:**

NA

---

> ### Author Response · Authors · 2024-11-20
>
> We would like to thank the reviewer for all the comments! Below we address the comments in the **Weaknesses** section.
> - Thanks for the suggestion on more experiments! In fact, this can already be observed from the existing experimental results: for the objective function in Eq.13, our online-PGM is a slight variation to RHIG and achieves similar theoretical performance. Thus, the comparison in Figure 4 can be viewed as between RHIG and online-AGM. In Figure 4, notice that when $k$ (the number of steps of prediction) increases, the regret of online-AGM converges to a smaller value than that of online-PGM. As pointed out in line 527 - 529, this might be explained by the robustness of online-AGM to long-term prediction error.
>
> - We use superscript to denote the number of updates that have been applied to a variable, and we use subscript to denote different decision variables. For instance, $x_s^{(l)}$ is the decision variable $x_s$ after $l$ steps of updates. For the online problem, our algorithms indeed construct the decisions in a timely manner ($x_t$ is decided at time $t$): our online-PGM takes $x_t$ as a weighted average of $x_t^{(1)},\ldots, x_t^{(L)}$, and our online-AGM takes $x_t = x_t^{(L)}$, where $L$ is the number of updates to each variable. This is online-implementable, since by time $t$,  $x_t^{(1)},\ldots, x_t^{(L)}$ have been calculated already.
> As an example, if $L=2$, and the graph is a line such that for all $t$, the neighbors for $x_t$ are $x_{t-1}$ and $x_{t+1}$, then $x_t^{(2)}$ depends on $x_{(t-1):(t+1)}^{(1)}$, which further depends on $x_{(t-2):(t+2)}^{(0)}$. Since $x_{(t-2):(t+2)}^{(0)}$ is available at time $t$, $x_{(t-1):(t+1)}^{(1)}$ can be computed and thus is available at time $t$. As a result, $x_t^{(2)}$ can be computed at time $t$, and so is the output.
>
> - When $(a,b) = (1,0)$, our online-AGM has better performance than previously proposed algorithms such as RHIG, in terms of dependency on the initial regret and robustness against long-term prediction error. In terms of the optimality, in Li \& Li (2020), a regret lower bound of $\Omega(\sum_{t=1}^T (\frac{1-\sqrt{\kappa}}{1+\sqrt{\kappa}})^{t-1}\delta_t)$ is proved, where the factor for the $t$-step prediction error is $\Omega(\frac{1-\sqrt{\kappa}}{1+\sqrt{\kappa}})^{t-1})$, but the corresponding factor for their proposed RHIG is $(1-\kappa/4)^{t-1}$. For our online-AGM, the corresponding factor is $(1-O(\sqrt{\kappa}))^{t-1}$, which achieves an optimal dependency on $\kappa$, thus closing this gap.

---

> > ### Author Response · Authors · 2024-11-25
> >
> > Dear reviewer, we again thank you for all the valuable feedbacks! Since the end of the discussion period is approaching, we would like to know if you have additional questions and/or concerns for our responses and our paper? We would be happy to provide further details and clarifications you need!

---

### Official Review · Reviewer_TE31 · 2024-11-04

**Soundness:** 3
**Presentation:** 2
**Contribution:** 2
**Rating:** 5
**Confidence:** 4

**Summary:**

This paper studies online convex optimization with predictions. The prediction means that at each time step t, predictions about the next k steps are available to the decision maker. The cost is a (strongly) convex function that not only depends on the current decision, but also depends on the actions chosen in the past and in the future. Based on the question proposed and studied in Li & Li (2020) and on the algorithms designed in (Li & Li, 2020; Li et al., 2021), this papers proposes an algorithm to solve this problem, and they show that the dynamic regret of the algorithm can be reduced by a exponentially decayed factor through accelerated gradient descent, at a cost of an additive error term that depends on the prediction accuracy.

**Strengths:**

1. Considering predictions in online convex optimization is interesting and important.
2. The proposed algorithm in this paper extends existing results in several meaningful directions.
3.  Proofs of the theoretical results are provided, and numerical experiments support the author's claims.

**Weaknesses:**

1. Although the paper mainly focuses on the theory, could you provide more concrete examples for the meaning of predictions in online convex optimization settings? Specifically, you could 1-2 specific real-world examples of how predictions could be used in online optimization problems, such as in energy systems or supply chain management.
2. In my opinion, it does not seem to be practical for the present cost function to be related to future decisions, since the future decision is a time-varying variable that can be independent of the decision in the present moment. In general, the decision maker will only make the current moment decision $x_t$ at time t. Could you please give a practical example detailing how the current cost function relates to future decisions?
3. About the parameter $\theta$: It seems that this paper considers the full information case rather than the bandit information case in online convex optimization, since both the cost function f_t and its gradient are used in the proposed algorithm. Regarding this point, could you clarify that after predicting $\theta$, is the specific expression of the cost function f_t() revealed to the decision maker or not? I suggest that the authors explicitly state in Section 1.1 whether the full cost function f_t is revealed after predicting θ, or if only gradient information is available.
4. In Li et al. TAC 2021, there is an initialization step in their algorithms that can effectively reduce the initialization error ||x^(init)-x^*||. The initialization step together with the gradient update step in their algorithms yield the overall regret bound of the algorithms. In this paper, the regret bounds of the proposed algorithm directly depend on R_0=||x^(init)-x^*|| which can scale as O(T) and in turn makes the overall regret bounds large. Could you comment on this point regarding why R_0 shows up in the regret bounds?
5. Compared to the existing papers, could you please describe/summarize the major novelty/technical difficulties in the algorithm design and regret analysis in this paper? In particular, it seems that the regret analysis follows closely from those in Devolder et al 2013a,b.
6. All the text sizes in the figures/algorithms/tables are too small; please try to improve the presentation/display of the relevant content.

**Questions:**

See weaknesses.

---

> ### Author Response · Authors · 2024-11-20
>
> Thank you for all the valuable comments and questions! We would like to provide the following clarifications to the concerns and questions raised in the **Weaknesses** section.
>
> 1. Yes, we will provide specific real-world examples in the revised version. Below are two possible relevant settings.
> - Consider a simplified data center management problem, where $\theta_t$ and $x_t$ are the numbers of requests and active servers at time $t$, respectively. Then $f_t$ is the sum of the operating cost (which depends on $x_t,\theta_t$) and the switching cost (which depends on $x_t - x_{t-1}$). Prediction in this setting means prediction of the number of requests during time $t+1,\ldots,t+k$.
> - Consider a quadrotor tracking problem, where $x_t$ and $\theta_t$ are the positions of the quadrotor and the target at time $t$, respectively. Then $f_t$ is the sum of the distance to the target ($\|x_t - \theta_t\|^2$) and the cost to change the position (which depends on $x_t - x_{t-1}$) and/or velocity (which depends on $(x_{t+1} - x_t)- (x_t - x_{t-1})$). Prediction in this setting means prediction of the positions of the target during time $t+1,\ldots,t+k$.
>
> 2. Indeed, the decision maker only decides $x_t$ at time $t$. However, the goal is to minimize the *cumulative cost* $C = \sum_{t=1}^T f_t$, which is fully determined by time $T$.
> - One practical example is tracking problems which also control the *acceleration*. Consider a quadrotor tracking problem, where $x_t$ and $\theta_t$ are the positions of the quadrotor and the target at time $t$, respectively. Noticing that $(x_{t+1} - x_t)- (x_t - x_{t-1})$ represents the acceleration, then $f_t(x_{(t-1):(t+1)};\theta_t) = (x_t - \theta_t)^2 + ((x_{t+1} - x_t)- (x_t - x_{t-1}))^2$ can be used to control the distance to the target together with the acceleration. In fact, this is an example of the ``aggregate information'' mentioned in motivating example 1, line 82 - 86.
> - Another practical example is delayed decision making (motivating example 2, line 87 - 90): if $f_t$ depends only on $x_{t+b}$ for all $t$, then $x_s$, the decision made at time $s$, affects only $f_{s-b}$. That is, the decision making is delayed for $b$ steps.
>
> 3. Sorry for the confusion about $\theta$. We assume that the functional form of $f_t(\cdot;\theta_t)$ is known for each $\theta_t$, and what is unknown is the true parameter $\theta_t^*$. We will clarify this in the setup.
>
> 4. Regarding $R_0$, we would like to make the following comments.
> - In Theorem 3 of Li et al. (2021), the authors give regret upper bound which depends on the regret of the initialization ($Reg(\phi)$ in their notation); in addition, this holds under the assumption that the objective function is strongly convex. Under this assumption, $\|\mathbf{x}^{(init)} - \mathbf{x}^{*}\|$ can be upper bounded by $Reg(\mathbf{x}^{(init)})$. Thus, as discussed in line 162 - 166, our Theorem 1.1 implies that the regret of our online-AGM is $O(\kappa^{-1}\rho_1^k\cdot Reg(\mathbf{x}^{(init)})) + \text{prediction error}$.
> - We also provide results when the strong-convexity assumption is relaxed ($\kappa=0$, Theorem 1.1). Without strong-convexity, indeed our regret depends on $R_0$. This inherits from the convergence analysis of classical offline algorithms such as (accelerated) gradient descent. To the best of our knowledge, this is the first performance guarantee for gradient-based algorithms for this problem without strong-convexity.
> - Admittedly, $R^2_0$ can scale as $O(T)$. However, when strong convexity holds, with even just $k = O(\log(T))$ steps of prediction and look-ahead initialization, constant (independent of $T$) regret can be achieved. Without strong-convexity, the regret for online-AGM is $O(T/k^2)$; if $k = \Omega(T^{\alpha})$ for some $\alpha>0$, then the regret is $O(T^{1-2\alpha})$, sublinear in $T$.
>
> 5. We believe that the main novelty is the connection between online and offline problems, as established in Section 3.2: we view offline iterative algorithms from a state-transition perspective, and thus can be implemented in an online fashion (exactly or approximately) using only sequentially revealed information. The performance of the online algorithms (regret) follows directly from the convergence guarantees of the offline algorithms, which greatly simplifies the analysis. In addition, the rich literature of classical offline optimization problems could bring new insights into the online optimization.
>
> 6. Thanks for the suggestion! We will improve the presentation/display in the revised paper.

---

> > ### Author Response · Authors · 2024-11-25
> >
> > Dear reviewer, we again thank you for all the valuable feedbacks! Since the end of the discussion period is approaching, we would like to know if you have additional questions and/or concerns for our responses and our paper? We would be happy to provide further details and clarifications you need!

---

> > > ### Comment · Reviewer_TE31 · 2024-11-28
> > >
> > > Thank the authors for preparing the response. I am still not fully convinced why the problem formulation is of practical importance and at the same time the technical contribution of this paper is also a bit limited. I will maintain my original score.

---

### Official Review · Reviewer_aW6q · 2024-11-04

**Soundness:** 3
**Presentation:** 2
**Contribution:** 2
**Rating:** 5
**Confidence:** 3

**Summary:**

The paper studies an online convex optimization setting in which
the losses at time $t$ may depend on both a history of previous decisions, as
well as future decisions, generalizing the standard OCO with memory setting.
Algorithms are designed for smooth+strongly-convex losses, as well as
for losses which are only strongly convex. Dynamic regret guarantees of the form
[initialization error]+[prediction errors] are proven. An online update scheme is
proposed to allow for more incremental computation.

**Strengths:**

- The proposed problem setting it is a very natural generalization of the usual OCO with memory setting.
- The results are novel and generalize those of prior works to a harder setting (access to only *inexact* predictions)

**Weaknesses:**

- Typically dynamic regret guarantees scale with some measure of how
much the comparator sequence varies, reflecting the "difficulty" of
a given sequence. Here no such term appears in the bounds, and instead there
are terms which involve prediction error of the future predictions given at the start of the round.
This seems not very meaningful to me; it essentially just says that the performance of the algorithm
will be roughly as good as the performance of the provided predictions. But there is no general strategy for
choosing a good sequence of these predictions, because at time $t$ the predictions $t+1,\ldots,t+b$ are chosen
at once, with no feedback. So the dynamic regret guarantee could be arbitrarily high,
even for "easy" comparator sequences.

- The draft suffers from several issues in writing quality
    -   The paper is incomplete, ending abruptly after the numerical experiments
        section with no conclusion / summary.

    - There are many instances of oddly structured, difficult-to-parse sentences
    (see, e.g., the first line of the abstract)

    -   The paper is somewhat oddly structured, with the main results preceeding
        the contributions section

- Figure 3 is pretty hard to follow, and the color coding is never really explained.
  Perhaps this would fit better in an appendix, wherein a proper concrete explanation of
  what's happening in the figure could be given. Ideally with some simple numerical values
  so the reader can more easily trace what's happening

-   Suggestion: the theory takes up the vast majority of the paper, and the experiments
    are not particularly interesting other than providing a sanity check that
    the algorithm works. I suggest positioning
    this work as a theory paper, and moving the experiments to the appendix entirely, so that
    there is room to provide a proper discussion of the results and conclusion at the end of the paper.

**Questions:**

- At time $t$, the learner suffers a loss which depends on $[x_{t-a},\ldots,x_t, \ldots,x_{t+b}]$,
and also receives feedback which is a function of these values. This implies that the learner must either commit to
$x_{t+1},\ldots, x_{t+b}$ ahead of time, on round $t$, or is allowed to rescind their choice and change their
choice of these variables on the next round. Which protocol is being used exactly?

---

> ### Author Response · Authors · 2024-11-20
>
> Thank you for the valuable feedback! We would like to make the following clarifications.
>
> ## Weaknesses
>
> - Regarding the *dynamic regret*, in our setup, the decisions at different times lie in different sets: $x_t\in \mathcal X_t$ and $\mathcal X_t$'s can be different for different $t$. For instance, if $\mathcal X_1 = [0,1]^{10}$ and $\mathcal X_2 = [0,1]^{20}$, then there is no natural way to quantify the variation between $x_1$ and $x_2$. Thus, we are not sure if the variation of the comparator sequence (or the offline optimal solution) is a good measure of the difficulty in our setup. If such measure of variation is available (e.g. $\mathcal X_t$'s are the same for all $t$), then the regret of our algorithms may depend -- implicitly through the regret of the initialization -- on the difficulty of the comparator sequence. For instance, if $a+b = 1$ and there are $k$-step exact predictions, then the regret (from line 165) of online-AGM is $O(\kappa^{-1}\rho_1^k\cdot Reg(\mathbf x^{(init)}))$. Thus, if $Reg(\mathbf x^{(init)}) = O(\sqrt{T\mathcal P_T})$ where $\mathcal P_T$ quantifies the variation of the comparator sequence, then the regret of online-AGM is $O(\kappa^{-1}\rho_1^k\cdot \sqrt{T\mathcal P_T})$.
>
> - Regarding the *writing quality*, we would like to thank the reviewer for all the suggestions! We will add a summary section and make the paper more readable by polishing and restructuring.
>
> - Regarding *Figure 3*, we use colors (degrees of ``redness'') to represent the number of updates that have been applied to the vertices: the darker a vertex is, the more updates it has received. We agree that Figure 3 is a bit confusing and need more explanations, and it might be better to move it to the appendix.
>
> - The suggestion that the experiments could be moved to the appendix sounds reasonable, and we will add a discussion/conclusion in the revision.
>
>
>
> ## Questions
> Sorry for the confusion. We assume that the functional form of $f_t(\cdot;\theta_t)$ is known for each $\theta_t$, and what is unknown is the true parameter $\theta_t^*$. The feedback is not the function values (evaluated at, say, $x_{t-a:t}$ and some hypothetical $x_{t+1:t+b}$); instead, after committing $x_t$ at time $t$ (and cannot be modified in the future), the process goes to time $t+1$, and at the beginning of time $t+1$, the decision maker is given information about $\theta_{t+1}^*$ (as well as about future parameters).

---

> > ### Author Response · Authors · 2024-11-25
> >
> > Dear reviewer, we again thank you for all the valuable feedbacks! Since the end of the discussion period is approaching, we would like to know if you have additional questions and/or concerns for our responses and our paper? We would be happy to provide further details and clarifications you need!

---

> > ### Comment · Reviewer_aW6q · 2024-11-25
> >
> > I don't think I understand the example; why would the decision set change from a 10-dimensional to a 20-dimensional problem on a different round? What type of problem is this supposed to be modelling exactly? This does not seem to be an issue that arises in any of the motivating examples given in the introduction. In the first motivating example a switching cost is even mentioned directly, suggesting that there is some meaningful way to measure the differences between decisions made on different rounds

---

> > > ### Author Response · Authors · 2024-11-26
> > >
> > > Thank the reviewer for the feedback!
> > >
> > > - Consider a 2D tracking problem, where $\mathbf x_t$ is a vector representing the positions of all the trackers. At time $t=1$, there are 5 trackers and so $\mathcal X_1\subset \mathbb R^{10}$; at time $t = 2$, there are 5 new trackers, and so the total number of trackers becomes 10, making the space $\mathcal X_2\subset \mathbb R^{20}$.
> > >
> > > - We do not assume that all $\mathcal X_t$'s are the same. However, if that is the case and there is a meaningful way to quantify the variation in the comparator sequence, then the regret of our algorithms may depend on the difficulty of the comparator sequence through the regret of the initialization.

---

> > > > ### Comment · Reviewer_aW6q · 2024-11-26
> > > >
> > > > I'm not fully convinced that the given example is really a problem since you've either defined a problem where learning is impossible (e.g. even against a fixed comparator no algorithm will be able to achieve low regret if you add a new decision variable on each round) or one could instead consider an equivalent problem where the shared fixed decision set is $\arg\max\|\mathcal{X}\|$, and the losses ignore specific subsets of the variables, in which case the usual comparator complexity penalty would be well defined, no?
> > > >
> > > > In any case, let's focus on the shared decision space setting for simplicity, since my point still seems to stand in this special case: the bounds presented here scale with the prediction error of the future predictions, which can be arbitrarily high even in easy problems, since there is no meaningful way to choose these a priori without further assumptions. So even if the regret of the initialization provides the measure of comparator complexity, the bound no longer has the property that the regret is necessarily smaller against "easy sequences"

---

> > > > > ### Author Response · Authors · 2024-11-26
> > > > >
> > > > > Indeed, the setting we study is general enough to model the case when learning is impossible, and we show that when learning is impossible, *prediction* can be very useful in regret minimization.
> > > > >
> > > > > When learning is possible, by choosing an initialization whose regret depends on the variation of the comparator sequence, our regret may depend on such variation (not on $T$). However, we believe that the dependence on the prediction error is unavoidable: in [1] (Theorem 3), the authors give a regret lower bound $\Omega(\sum_{t=1}^T \rho^{t-1}\delta_t)$ for $\rho = (\frac{1-\sqrt{\kappa}}{1+\sqrt{\kappa}})^{2}$; however, the corresponding factor for their algorithm RHIG is $\rho = 1-\kappa/4$. In online-AGM, the factor is $\rho = 1-O(\sqrt{\kappa})$, which has the correct dependence on $\kappa$.
> > > > >
> > > > > [1] Li, Y., & Li, N. (2020). Leveraging predictions in smoothed online convex optimization via gradient-based algorithms. Advances in Neural Information Processing Systems, 33, 14520-14531.

---

### Official Review · Reviewer_Hi6C · 2024-11-04

**Soundness:** 2
**Presentation:** 1
**Contribution:** 2
**Rating:** 3
**Confidence:** 4

**Summary:**

This paper studies the problem of online convex optimization with predictions, where predictions about the next $k$ steps are available and the costs are coupled over time steps. Specifically, the authors propose two algorithms, namely online-PGM and online-AGM. Compared with the existing results, the author's analysis implies that their online-AGM algorithm can reduce a part of the dynamic regret when loss functions are strongly convex and smooth. Moreover, if the functions are not strongly convex, the authors also establish a theoretical guarantee on the dynamic regret bound of the proposed algorithms.

**Strengths:**

1) Two new algorithms, especially online-AGM, are developed for the problem of online convex optimization with predictions and coupled cost.
2) By utilizing some nice properties of the accelerated gradient descent method, the proposed online-AGM algorithm can improve the dynamic regret bound established by previous studies (note that the existing result holds only for a special case of the problem studied in this paper).
3) The authors also establish dynamic regret bounds for their algorithms in the case with smooth but not strongly convex functions.

**Weaknesses:**

1) The studied problem seems to be very complicated but is not well-motivated. The three motivating examples discussed in this paper are not convincing enough. Specifically, for the first example, it is unclear what applications the so-called aggregate information must be considered. For the second example, although the delayed decision making can be regarded as a special case of the studied problem, the delayed problem itself has been well studied, and it is not necessary to consider a more complicated setting unless some new insights can be derived. For the third example, its connection to online applications is unclear, and I especially cannot understand why it is related to online learning with prediction.
2) The contributions of this paper seem to be incremental, and actually are not described clearly. First, although two algorithms are proposed and analyzed in this paper, it seems that online-AGM is better than online-PGM in both theoretical guarantees and the experimental results. As a result, the significance of online-PGM seems to be questionable. Second, as discussed in Sections 1.2 and 1.3, the main theoretical improvement of this paper is achieved by the online-AGM algorithm, and limited to one of the two components of the dynamic regret bounds. More specifically, it seems that such an improvement is mainly achieved by refining the initialization by utilizing the accelerated gradient descent method, whose novelty is limited.
3) The assumptions on loss functions and predictions are too strong, which will limit the application of the proposed algorithms. Moreover, some assumptions are not described clearly. For example, according to the abstract, only the predictions about the next $k$ steps are available. However, according to lines 72 to 63, it seems that the predictions about all the future steps should be available.

**Questions:**

Besides the weaknesses discussed above, I am also concerned about the writing of this paper and would like to provide some suggestions in the following.
1) The latex environment utilized for *Figure* should be checked. For example, in this paper, the captions for Figures 1, 2, and 3 are presented above these figures.
2) The authors should first discuss the technical challenges of minimizing the dynamic regret of the problem studied in this paper, and provide more technical details of existing algorithms, which is helpful to understand the technical novelty of the proposed algorithm.
3) The authors should clearly explain whether the proofs of Theorems B.1 and B.2 belong to the contributions of this paper. If the answer is not, a simple reference to previous studies is sufficient for utilizing these results.
4) The meaning of $\kappa$ is unclear when we read the abstract, and the meaning of $L$ is unclear when we read Theorem 1.1.

---

> ### Author Response · Authors · 2024-11-20
>
> Thank you for all the valuable comments and questions. We would like to make the following comments.
> ## Weaknesses:
> 1. Regarding the motivation and setup, we would like to provide the following details for the three motivating examples.
> - For ``aggregate information'', consider a quadrotor tracking problem, where $x_t$ and $\theta_t$ are the positions of the quadrotor and the target at time $t$, respectively. Then, the difference $x_t - x_{t-1}$ represents the velocity, the second order difference $(x_{t+1} - x_t)- (x_t - x_{t-1})$ represents the acceleration, and the moving average $\frac{1}{a+b+1}\sum_{s = t-a}^{t+b} x_s$ represents the average position of the quadrotor between time $t-a$ and time $t+b$. Thus, when one wants to control the velocities, accelerations, and/or average positions of the quadrotor, the natural way to model the problem is to assume that $f_t$, the objective function at time $t$, depends on $x_{(t-a):(t+b)}$.
> - Previous works on online optimization with delayed feedback usually study the static regret or competitive ratio (see [1][2] and the references therein), whereas we look at the dynamic regret; in addition, our work provides some insights into the benefits of having predictions in online decision making.
> - For the third motivating example, consider the following problem: there are $B$ servers; at day $t=1,2,\ldots,T-a$, one request arrives and $c_t$, its total reward, is revealed. The request takes $(a+1)$ days to be fulfilled, and can be fulfilled partially: if $0\leq x\leq 1$ fraction of it is fulfilled, the reward will be $c_tx$, and this requires $x$ server from day $t$ till day $t+a$. The decision maker needs to decide $x_t$, the fraction of the day-$t$ request fulfilled, at time $t$, with the goal of maximizing the total reward. (In this case, $b=0$.)
>
>     This can be formulated as an online linear programming problem: $$\min_{x_t\in [0,1],~ t = 1,\ldots,T-a} -\sum_{t=1}^{T-a} c_tx_t,\quad s.t.~\sum_{t=\max(1,s-a)}^{s} x_t\leq B,\quad s = 1,\ldots,T-a.$$
>   The Lagrangian for this problem is
>     $$\mathcal L(x_{1:(T-a)}, \lambda_{1:(T-a)}) = -\sum_{t=1}^{T-a} c_tx_t + \sum_{s = 1}^{T-a} \lambda_s (\sum_{t=\max(1,s-a)}^{s} x_t- B),$$
>     where $\lambda_s$'s are dual variables and can be interpreted as the shadow prices of the (budget) constraints. Due to strong duality of LP, the original constrained LP is equivalent to minimizing $\mathcal L(x_{1:(T-a)}, \lambda^*_{1:(T-a)})$ where $\lambda^*_{1:(T-a)}$ is an optimal dual. In this setting, one can take
>     $$f_t(x_{\max(1,t-a):t};\theta_t) :=  - c_tx_t +  \lambda_t (\sum_{s=\max(1,t-a)}^{t} x_s- B),\quad \theta_t = (c_t,\lambda_t),$$
>     with the true parameter $\theta_t^* = (c_t,\lambda_t^*)$. Then, prior information about the sequence $(c_1,\ldots,c_{T-a})$, such as its distribution, might be used to predict $(\theta_1^*,\ldots,\theta_{T-a}^*)$.
>
> [1] Yu-Guan Hsieh, Franck Iutzeler, Jérôme Malick, and Panayotis Mertikopoulos. 2022. Multi-agent online optimization with delays: asynchronicity, adaptivity, and optimism. J. Mach. Learn. Res. 23, 1, Article 78 (January 2022), 49 pages.
>
> [2] Weici Pan, Guanya Shi, Yiheng Lin, and Adam Wierman. 2022. Online Optimization with Feedback Delay and Nonlinear Switching Cost. Proc. ACM Meas. Anal. Comput. Syst. 6, 1, Article 17 (March 2022), 34 pages. https://doi.org/10.1145/3508037.
>
> 2. Regarding contributions, we would like to make the following clarifications.
> - Indeed, online-AGM has better theoretical and experimental performance than online-PGM, and we believe our main algorithmic contribution is online-AGM. We provide the results for online-PGM as a comparison to show the benefit of acceleration. In addition, in the special setting of smoothed online convex optimization studied in Li \& Li (2020), our online-PGM is similar to their RHIG. Moreover, we also provide regret guarantees when strong-convexity is relaxed for both online-PGM and online-AGM. To the best of our knowledge, this is the first theoretical guarantee for gradient-based algorithms without strong-convexity in this line of work of online convex optimization with prediction.
>
> - Our online-AGM improves both components in the dynamic regret. For the initialization regret, we improve the factor from $(1-\kappa/4)^{k}$ to $(1-O(\sqrt{\kappa}))^{k}$. For the dependency on the $l$-step prediction error, we improve the factor from $(1-\kappa/4)^{l}$ to $(1-O(\sqrt{\kappa}))^{l}$ -- this implies that our online-AGM is also more robust to long-term prediction error. In fact, Li \& Li (2020) shows that the regret's dependency on $l$-step prediction error is $\Omega((\frac{1-\sqrt{\kappa}}{1+\sqrt{\kappa}}))^{l})$, yet their RHIG achieves only $(1-\kappa/4)^{l}$. Thus, in terms of the dependency on long-term prediction error, our online-AGM has an optimal dependency on $\kappa$, and closes the gap. We will expand the discussion in line 178 - 185 to make this more clear.

---

> > ### Author Response · Authors · 2024-11-20
> >
> > 3. Regarding the assumptions,
> > - Assumption 1.1: $\nabla f$ is Lipschitz in $\theta$. This assumption is satisfied by many applications of interest. For instance, $\|x_t - \theta_t^*\|^2$ which is common in tracking problems ($\theta_t^*$ is the target position at time $t$). Another example is $f_t(x;\theta_t) := E[F_t(x;Z)]$ where $\theta_t$ represents a distribution over some set $\mathcal{Z}$, $Z\sim \theta_t$, and $\|\nabla F_t(x;z)\|\leq M$ for all $x,z$, then $\nabla f_t$ is $M$-Lipschitz in $\theta_t$ (w.r.t. the total variation distance).
> > - Assumption 1.2: $C$ is (strongly) convex and smooth. These assumptions are inherited from the assumptions in the analysis of offline (classical) PGM and AGM.
> > - Assumptions on predictions. We apologize that the description of the prediction in the abstract and line 72 - 73 are confusing. Indeed, in line 72 - 73, we assume that at each time $t$, there is prediction for the entire sequence $\boldsymbol{\theta}^*$, denoted as $\widehat{\boldsymbol{\theta}}^{(t)}$. Nevertheless, the implementation of the algorithms (online-AGM and online-PGM, with $L$ steps for some $(a+b)L\leq k$), at time $t$, depends only on the $t-k$ to $t+k$ components of $\widehat{\boldsymbol{\theta}}^{(t)}$. In addition, in Theorem 1.1 line 157 - 160, the regret depends only on the prediction accuracy for $\widehat{\theta}^{(t)}_{(t-k):(t+k)}$ -- in particular, the prediction accuracy for $\widehat{\theta}^{(t)}_s$ for $s>t+k$ does not matter. Thus, if one has $k$
> > -step prediction for the parameter sequence, one can take $\widehat{\theta}^{(t)}_s$ arbitrarily for $s>t+k$, without affecting the algorithms or the regrets. We will clarify this in the revised paper.
> >
> >
> > ## Questions
> > Thank you for the suggestions! We will check the latex environment for figures, add discussions on the challenges of dynamic regret and details of existing algorithms, and clarify $\kappa$ and $L$. For Theorem B.1 and B.2, as pointed out in line 727 - 731, they are adapted from Devolder et al. (2013a,b), taking into account the variation of the oracle errors across time. Specifically, Devolder et al. (2013b) studies smooth, convex but not strongly convex optimization with inexact first oracle; Devolder et al. (2013a) studies the strongly convex case, where the inexactness ($\delta_t$) of the first order oracles is the same for all time period. In Theorem B.1 and B.2, we explicate the dependence of suboptimality on $\delta_t$ for all $t$ under strong-convexity. That is, we provide convergence guarantees for strongly convex case, with time-dependent inexactness $\delta_t$.

---

> > > ### Author Response · Authors · 2024-11-25
> > >
> > > Dear reviewer, we again thank you for all the valuable feedbacks! Since the end of the discussion period is approaching, we would like to know if you have additional questions and/or concerns for our responses and our paper? We would be happy to provide further details and clarifications you need!

---

> > > > ### Comment · Reviewer_Hi6C · 2024-11-28
> > > >
> > > > Sorry for the late reply. I have read your responses, and most explanations in the responses are helpful for improving your manuscript. However, the additional explanations about motivating examples are still not very convincing. For example, although the velocities, accelerations, and/or average positions of the quadrotor may be controlled at the same time in the quadrotor tracking problem, it is not a good way to model them in a single function because this way may fail to ensure an accurate control on each of these metrics. For online optimization with delayed feedback, there already exist previous studies that consider dynamic regret. Moreover, although both components in the dynamic regret have been improved, the technical novelty of this paper seems to be still limited to refining the initialization (via the accelerated gradient descent). Additionally, I believe that this paper needs a major revision to address the writing problem, which is a common concern of most reviewers. For these reasons, I maintain my original score.

---

### Meta-Review · Area_Chair_azKy · 2024-12-11

**Metareview:**

This paper investigates online convex optimization (OCO) with predictions, extending prior work by introducing a framework where the cost at each time step depends on both past and future decisions. The authors propose two algorithms, online-PGM and online-AGM, showing that the latter improves dynamic regret, particularly in its robustness to long-term prediction errors.

While the paper generalizes previous work and provides a more robust solution to prediction errors, its novelty is questioned by some reviewers, as it relies heavily on existing methods. Although the experiments support the theoretical claims, the practical relevance of the work remains unclear to some reviewers, and the connection to traditional online prediction settings is disputed. Additionally, concerns regarding the clarity and structure of the writing were raised.

**Additional Comments On Reviewer Discussion:**

The authors' responses addressed some concerns. However, some issues remain unresolved. Reviewers are still unconvinced about the practical relevance of the problem and feel the technical contributions are incremental, with limited novelty compared to existing work.

---

### Decision · Program_Chairs · 2025-01-22

Reject